# SELECT: SELEctive Context Transfer *for* Class-Incremental Semantic Segmentation

## Abstract

Class-Incremental Semantic Segmentation (CISS) is fundamentally challenged by catastrophic forgetting and background shift, where learning new concepts degrades performance on previously seen classes. While existing methods attempt to balance stability (retaining old knowledge) and plasticity (learning new knowledge), they often fail to leverage prior knowledge effectively. These approaches typically rely on indiscriminate knowledge transfer or ambiguous initializations, which can dilute crucial semantic information. To overcome this limitation, we propose **SELECT**, a novel approach for **Sele**ctive **C**ontext **T**ransfer. SELECT intelligently transfers knowledge by focusing on inter-class semantic similarity. Its core is a Context Transfer Attention mechanism that identifies semantically related past classes and selectively transfers their contextual information to guide the learning of new ones. To ensure this transfer does not corrupt the original representations, we introduce a controlled perturbation and novel discriminative loss that explicitly enforces separation between the latent spaces of the new class and its influential old-class counterparts. Extensive experiments on PASCAL VOC 2012 and ADE20K demonstrate that SELECT significantly mitigates catastrophic forgetting and background shift, providing a robust and effective solution to the stability-plasticity dilemma.

## 1 Introduction

Semantic segmentation, the task of assigning a class label to every pixel in an image Strudel et al. (2021); Zhang et al. (2022a); Xie et al. (2021), is a cornerstone of modern computer vision. Its success underpins critical applications, from autonomous vehicles navigating complex urban scenes to medical imaging systems identifying pathological tissues Li et al. (2023); Mourdi et al. (2024). However, conventional neural network-based models are trained in a static, "closed-world" setting, assuming all classes are known beforehand. This paradigm fails in real-world scenarios, which are inherently dynamic–an autonomous system must continually learn to adapt to novel road conditions and unpredictable traffic; and a diagnostic tool must adapt to novel disease manifestations. This necessitates a shift towards Class-Incremental Semantic Segmentation (CISS), where networks must continually learn new classes without needing to be retrained from scratch on all accumulated data. The primary obstacle in CISS is the well-known phenomenon of catastrophic forgetting Li & Hoiem (2018); Kirkpatrick et al. (2017); Cao et al. (2024), where the acquisition of new knowledge drastically degrades performance on previously learned classes. Therefore, developing a truly effective CISS solution requires navigating a challenging dilemma of competing objectives. First, it must maintain ① *Stability*, ensuring that knowledge of past classes remains intact. Second, it needs ② *Plasticity*, the ability to effectively learn and adapt to new classes. Finally, it must facilitate efficient ③ *Knowledge Transfer*, leveraging relevant prior knowledge to accelerate the learning of new, related concepts while maintaining both stability and plasticity. It must therefore find a principled equilibrium among these challenges.

Unfortunately, existing CISS methods struggle to achieve this balance, often relying on flawed heuristics. In these methods, a common strategy has been to initialize new class classifiers using information from the "background" class Cermelli et al. (2020); Cha et al. (2021). This approach may be suboptimal, as the background is a semantic mixture—a collection of ambiguity containing everything the model has learned not to recognise. We show the impact of this approach in Fig. 1, where class token distribution before and after remains largely unchanged. Initializing a specific

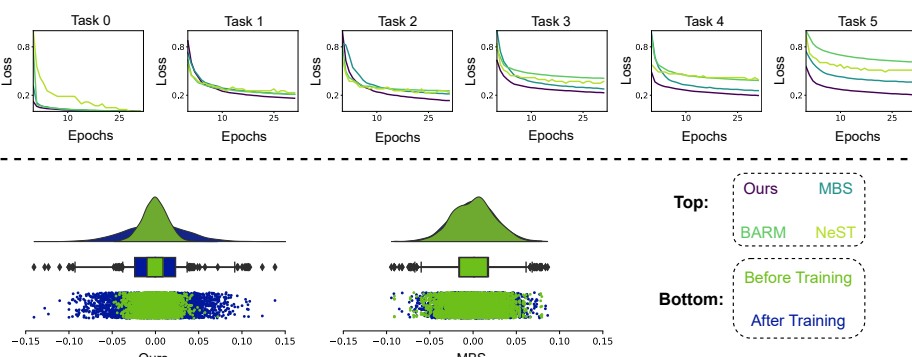

Figure 1: **[Top]***Faster Convergence:* Experimented on Pascal VOC $15 - 1$, our SELECT approach consistently achieves a lower training loss, and as the incremental tasks increase, we have a steeper loss curve, suggesting faster convergence. **[Bottom]** *Better Initialization:* Our method's initial token distribution has low variance suggesting that selective initialization is cohesive and precise before training. After training, it becomes more distinct and discriminative. Our method starts with a focused prior and learns to create a "buffer zone" to avoid confusion with other classes. In contrast, Background initialization used in MBS starts with a more ambiguous, higher-variance initialization which remains unchanged after training.

new category from this mixture provides a poor inductive bias, disturbing plasticity from the outset. Goswami et al. (2023) proposes using relevant information from the old background ; however, it incurs significant memory overhead. Acknowledging the need for better priors through semantic relatedness Ke et al. (2020); Hiratani (2024), other methods attempt to transfer knowledge from all past classes, often through global knowledge distillation Xie et al. (2024). While well-intentioned, this strategy dilutes the knowledge from a few relevant classes with overwhelming noise from many irrelevant ones (further discussed in §3.2).

To overcome these limitations, we argue that an optimal CISS framework must pivot from indiscriminate heuristics to a strategy of guided, adaptive knowledge transfer. We introduce *SELECT* (SELEctive Context Transfer), a novel approach that operates on this principle. SELECT's first step is to identify a small subset of past classes, $\mathcal{C}_s$, that are semantically most relevant to the new classes. This selective focus directly counters the noise and dilution problems of prior work. To execute the transfer, we introduce a **C**ontext **T**ransfer **A**ttention (**CTA**) mechanism. CTA acts as a high-precision instrument, selectively modulating and distilling only the most useful contextual information from the source classes in $\mathcal{C}_s$ to inform the learning of the new ones. This provides a powerful and clean semantic initialization, directly promoting both plasticity and effective knowledge transfer.

Crucially, this useful knowledge transfer must not come at the cost of forgetting prior information. The very act of linking new classes to old ones risks distorting the source information Shang et al. (2023). To address this, we introduce a two-part solution. Firstly, we introduce a controlled noise in the context transfer attention. This perturbation regularizes the information (through newly initialized tokens) and creates a small gap for adapting to a new class while also maintaining alignment to the previous knowledge. Further, we also introduce a dedicated context transfer loss. This loss function acts as a protective measure, explicitly enforcing a margin in the latent space between the new class information and those of their source classes in $\mathcal{C}_s$. This prevents information overlap and preserves the integrity of established knowledge. Thus, SELECT creates a symbiotic learning process: CTA promotes plasticity through targeted transfer, while the dedicated loss aids stability, resolving the core problem of stability-plasticity dilemma in CISS. We observe the result of this approach when observing the distribution of token values before and after training. In Fig 1, initial token distribution is precise and cohesive which changes to more discriminative after training. Also, this consequently results in faster convergence as tasks increase with time.

Our key contributions are summarized as follows: ❶ We propose a novel selective context transfer strategy that identifies a subset of semantically relevant classes from prior tasks to facilitate selective guidance to new classes. ❷ We introduce Context Transfer Attention (CTA), a mechanism for effectively and selectively transferring useful knowledge from similar past classes to the current class representation. ❸ We propose a context transfer loss to encourage distinct information between

previously learned similar classes and the transferred knowledge used for new classes, ensuring guided stability. ❹ We provide an empirical evaluation on the Pascal VOC and ADE20K datasets demonstrating the effectiveness of SELECT. Rather than prioritizing achieving peak benchmark performances, this work focuses on proposing a selectively adaptive knowledge transfer framework that enables a clearer understanding of incremental adaptation.

## 2 RELATED WORKS

### 2.1 INCREMENTAL LEARNING

Incremental learning Wang et al. (2025); Elsayed & Mahmood (2024); Kim et al. (2024b) has attracted growing attention for its ability to learn new knowledge without catastrophically forgetting the past. Li & Hoiem (2017) proposed the technique of knowledge distillation to deal with catastrophic forgetting, while Kirkpatrick et al. (2017) proposed the idea of selectively updating the important network weights from the previous task. Some methods have used regularization-based approaches Huang et al. (2024); Yildirim et al. (2023) where the models introduce the regularization terms and often penalize changes in the parameters to approximate the similar output distribution on previous tasks as the old model. Another approach, like architectural-based methods Song et al. (2024); Bonato et al. (2024), aims at designing new networks that are adaptable to incremental tasks. Some methods use replay-based approaches Lim et al. (2024); Zhou et al. (2024); Lopez-Paz & Ranzato (2017); Chaudhry et al. (2018) to memorize a small set of training images or representations from previous tasks and utilize them while learning the new tasks. Some works have also leveraged generative-based approaches Kamra et al. (2017); Shin et al. (2017) to learn about new tasks using a mixed set of real images from new tasks and generated images from previous ones.

### 2.2 KNOWLEDGE TRANSFER IN INCREMENTAL LEARNING

Most existing works on incremental learning focus on the issue of catastrophic forgetting. Rosenfeld & Tsotsos (2018) proposed to optimize loss on new tasks using old tasks' weights. Knowledge transfer Lopez-Paz & Ranzato (2017); Chaudhry et al. (2018) focuses on transferring the knowledge from previously learned tasks. Previous works Zeng et al. (2019); Dhar et al. (2019); Ruvolo & Eaton (2013); Hiratani (2024) have established that knowledge transfer has helped the new task learn better. However, these works have used traditional machine learning methods like regression Ruvolo & Eaton (2013), where forgetting is minimal. Recently, Ke et al. (2020) focussed on learning the mixed sequence of similar and dissimilar tasks that dealt with forgetting and further improved learning.

### 2.3 CLASS-INCREMENTAL SEMANTIC SEGMENTATION

The advancements in class incremental learning (CIL) tasks encouraged us to move towards dense prediction tasks, like semantic segmentation Strudel et al. (2021); Zhang et al. (2022a); Xie et al. (2021), that are still susceptible to catastrophic forgetting. To alleviate such issues, Michieli & Zanuttigh (2019) proposed class-incremental semantic segmentation that aims to segment the new classes without forgetting the previously learned ones. MiB Cermelli et al. (2020) adapted using knowledge distillation and addressing the background shift problem. Following that, several works Michieli & Zanuttigh (2021); Zhu et al. (2024); Sun et al. (2024); Goswami et al. (2023); Gong et al. (2024); Cermelli et al. (2023); Kim et al. (2024a); Zhang et al. (2023); Zhao et al. (2022); Chen et al. (2024) have been introduced to mitigate this issue. PLOP Douillard et al. (2021) addressed this problem using multi-scale distillation and pseudo-labelling strategy, while Incrementer Shang et al. (2023) proposed a transformer-based architecture and learning only relevant previous tasks features. Park et al. (2024); Zhang & Gao (2024); Fang et al. (2025) integrated old and new task method parameters to address it. Several works adapted replay-based approaches by either accessing previous data or representations in the form of prompts Chen et al. (2023b) or generating a few samples Chen et al. (2023a) and learnt with the current task. Our work resembles Cermelli et al. (2020); Park et al. (2024); Zhang & Gao (2024), where we utilize the contrastive learning methodology to establish the difference between categories. However, our method differs in that we establish the contrast between the tasks as well as the categories within each task.

## 3 METHOD

### 3.1 PROBLEM FORMULATION

The objective is to train a single model that sequentially learns to segment new classes over a series of tasks without forgetting previously acquired knowledge. The learning process is structured as a sequence of tasks, indexed by $t \in \{0, 1, ..., T\}$. The initial task, $t = 0$, is the base task, while tasks with $t > 0$ are subsequent incremental tasks. Each task $t$ introduces a new, distinct set of semantic classes $\mathcal{C}_t$, such that $\mathcal{C}_i \cap C_j = \emptyset$ for any $i \neq j$. The training data available for task $t$ is the dataset $\mathcal{D}_t = (x_i, y_i)_{i=1}^{N_t}$, where input $x_i \in R^{H \times W \times 3}$ and their corresponding ground-truth $y_i$ contain labels only from the class set $\mathcal{C}_t$. $N_t$ denotes the total number of inputs in the task $t$. Let $f_t$ denote the encoder-decoder-based model Strudel et al. (2021) for task $t$.

At $t = 0$, $f_0$ is trained on an initial set of classes $\mathcal{C}_0$. Specifically, for an input $x_i$, we obtain the encoded feature representations as $z_{(enc,i)} = f_0^{enc}(x_i)$. Alongside these encoded representations, we introduce a set of learnable class tokens $e_0 \in \mathbb{R}^{C_0 \times d}$, where $d$ is the token dimension. These tokens, one for each class, are randomly initialized and optimized to capture the core semantic characteristics of their respective classes, which are then used by the decoder for segmentation: $z_{(dec,i)}, \hat{e}_{(0,i)} = f_0^{dec}(z_{(enc,i)}, e_0); \hat{y}_i = z_{(dec,i)} \otimes \hat{e}_{(0,i)}$, where $z_{(dec,i)}, \hat{e}_{0,i}$ are the decoded feature representations, and decoder-generated class embeddings, respectively. Here, $\hat{e}_{(0,i)} \in \mathbb{R}^{C_0 \times d}$ captures how each class interprets an image and $\hat{y}_i$ is the predicted mask.

At $t > 0$, the learning process is constrained: to learn the new model $f_t$, we only have access to the previous model state, $f_{t-1}$, and the current task's dataset, $\mathcal{D}_t$. The goal is to produce a model $f_t$ that performs well not only on the new classes $\mathcal{C}_t$ but also on the cumulative set of all classes seen so far, $\mathcal{C}_{\leq t} = \bigcup_{i=0}^{t} \mathcal{C}_i$. Specifically, when a new class $c \in \mathcal{C}_t$ is introduced, the framework first identifies previously learned classes in $\mathcal{C}_{0:t-1}$ that might be semantically similar. Knowledge is then selectively transferred from these related classes to construct an informed and guided initial representation for the new class. This transferred knowledge provides a strong inductive bias, grounding the new concept within the model's existing semantic space. Finally, this new representation is fine-tuned using the current task's data. The subsequent sections detail the core components of our method. An overview of the proposed **SELECT** framework is visualized in Fig. 3.

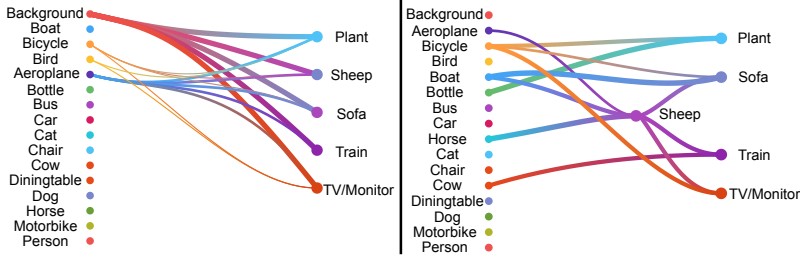

Figure 2: **[Left]** In NeST Xie et al. (2024), most of the initial knowledge for incremental classes is transferred from the background. **[Right]** Our proposed adaptive strategy focuses only on relevant classes.

### 3.2 ANALYSIS ON EFFICIENT KNOWLEDGE TRANSFER

In this section, we analyze the predominant knowledge transfer strategies in CISS and attempt to highlight the underlying limitations. Our analysis is grounded in two standard incremental settings on the Pascal VOC dataset, $15 - 1$ and $15 - 5$, under the overlapped scenario detailed in §4.1.

A popular heuristic in many prior works is to leverage the *background class* as the primary source of knowledge for new classes Cermelli et al. (2020); Park et al. (2024); Cha et al. (2021). The rationale is that the background representations serve as a knowledge base of all objects not yet explicitly learned. However, this base is an unstructured collection of features from countless unseen categories. Reeling on such a semantically mixed source for initialization can confuse the model and provide a poor inductive bias, rather than targeted guidance.

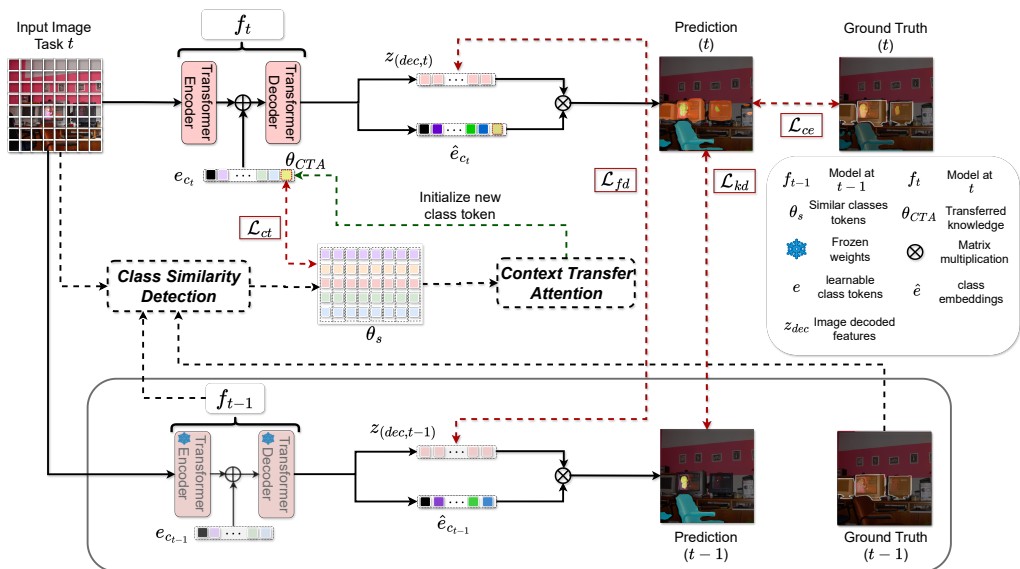

Figure 3: **SELEctive Context Transfer (SELECT).** The task $t$ dataset is processed by the previous task's model $f_{t-1}$ to identify the most similar class for each image using **Class Similarity Detection** (see §3.3). The similar classes' tokens $\theta_s$ are then processed by **Context Transfer Attention** (see §3.4) to formulate the adaptive token $\theta_{CTA}$ for the new class that is further used in the current task's model $f_t$. The model is trained using $\mathcal{L}_{ce}, \mathcal{L}_{fd}, \mathcal{L}_{kd}$, and the proposed context transfer loss $\mathcal{L}_{ct}$ enforcing distinction between $\theta_{CTA}$ and $\theta_s$ (see §3.5)
.

More recent methods have attempted to move beyond this simple heuristic. For instance, a strategy proposed by REMINDER Phan et al. (2022) primarily focused on to reduce the likelihood of forgetting prior similar classes, regardless of the new class initialization.

Another recent strategy proposed by Xie et al. (2024) involves fine-tuning the classifiers of all previously learned classes on the current task's data before transferring the weights. While this aims to identify relevant priors, we observe that the process is still dominated by non-specific features from the background class. As illustrated in Fig. 2, the resulting class still relies highly on drawing almost all its influence from the background.

Table 1: Performance comparison of NeST Xie et al. (2024) on Pascal VOC. † denotes results are reproduced. $bg$ denotes that only background initialization is used. ■ denotes the difference in the performance of both approaches.

| | 15-1 | | | 15-5 | | |
|---|---|---|---|---|---|---|
| | 0-15 | 16-20 | All | 0-15 | 16-20 | All |
| NeST† | 57.2 | 32.1 | 51.3 | 76.8 | 52.6 | 71.0 |
| NeST†$_{bg}$ | 65.6 | 34.6 | 58.2 | 75.2 | 49.0 | 68.9 |
| | -8.4 | -2.5 | -6.9 | +1.3 | +3.6 | +2.1 |

To rigorously test this observation, we conducted a controlled experiment. We compare the performance of the strategy from Xie et al. (2024) against a simplified ablation that uses only the background weights instead of the complete class matrix. The results, reported in Table 1, are interesting. The full method, despite its complexity, offers no significant advantage; its performance is only marginally different from the background-only baseline. This finding suggests that even this approach is may not be optimal. It strongly motivates the need for a new heuristic that enables targeted and selective knowledge transfer.

## 3.3 CLASS SIMILARITY DETECTION

In this section, we formalize the selection of a set of similar past classes, $\mathcal{C}_s \subseteq \mathcal{C}_{0:t-1}$, for a new task $t$. Our approach is based on a core intuition: *a previously learned class $k$ is semantically relevant to a new class if the established representation for $k$ exhibits stability when the previous model is exposed to images of the new class.* We calculate this stability by measuring the perturbation between a class's static learned token and its dynamically generated context-dependent representation (class embedding).

### 3.3.1 MEASURING REPRESENTATIONAL PERTURBATION

The model from the previous step, $f_{t-1}$, encapsulates the learned knowledge of all past classes $\mathcal{C}_{0:t-1}$. We use this frozen model as a probe. For each image-mask pair $(x_i, y_i)$ in the new dataset $\mathcal{D}_t$, we feed the masked image $x_i \odot y_i$ into $f_{t-1}$, where $\odot$ is the Hadamard product. For every past class $k \in \mathcal{C}_{0:t-1}$, we extract two distinct representations:

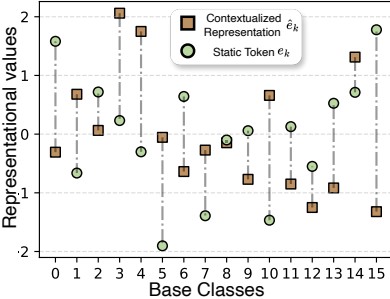

**Static Token** ($e_k$): This represents the canonical identity of the class $k$ and is independent of the current input.

**Contextualized Representation** ($\hat{e}_k^{x_i}$): The generated embedding for class $k$ produced by the model's encoder-decoder pipeline when processing the specific input $x_i$. This reflects how the model's understanding of class $k$ is influenced by the context of the new image.

Figure 4: Representational perturbation between static token $e_k$ and contextualized representation $\hat{e}_k$ (averaged).

A small deviation between these two vectors indicates that the content of the image $x_i$ is contextually aligned with class $k$. We measure this deviation, or perturbation, using the Euclidean distance:

$$\delta_{i,k} = \|\hat{e}_k^{x_i} - e_k\|_2 \tag{1}$$

As shown in Fig. 4, classes with strong semantic relevance exhibit significantly lower average perturbation across the new dataset. Similar plots targeting individual samples along with further analysis are provided in §C.

### 3.3.2 CONSENSUS-BASED SELECTION

To robustly identify the set of similar classes, we perform a consensus-gathering process over the entire dataset $\mathcal{D}_t$. First, for each image $x_i$, we identify the single past class $k_i$ that is mostly contextually aligned by finding the one with the minimum representational perturbation:

$$k_i = \arg \min_{k \in \mathcal{C}_{0:t-1}} \delta_{i,k} \tag{2}$$

This process yields a set of best-match candidates, $K = \{k_i | (x_i, y_i) \in \mathcal{D}_t\}$. We then agregate these votes by computing the frequency, $\tau$, for each unique past class $u \in \mathcal{C}_{0:t-1}$ within the candidate set $K$:

$$\tau = \sum_{k \in K} \mathbb{1}[k = u] \tag{3}$$

where $\mathbb{1}[\cdot]$ is the indicator function. Finally, to ensure that only classes with consistent and strong relevance are chosen, we select the final set of similar classes $\mathcal{C}_s$ by retaining classes whose frequency surpasses a threshold controlled by a hyperparameter $\varepsilon \in [0, 1]$. The resulting set $\mathcal{C}_s$, contains past classes that provide robust "positive contextual knowledge". This set is then used to initialize the new class representation as described in §3.4.

### 3.4 CONTEXT-LEARNED SELECTIVE KNOWLEDGE TRANSFER

Having identified the set of semantically similar source classes $\mathcal{C}_s$ using the method described in §3.3, we now detail the core mechanism for transferring their knowledge. The objective is to synthesize a potent initial class token for a new class by selectively aggregating learned tokens from these source classes. This is accomplished by our **C**ontext **T**ransfer **A**ttention (CTA) module. Let $\mathcal{C}_s (\subseteq \mathcal{C}_{0:t-1}) = \{c_1, c_2, ..., c_n\}$ be the set of $n$ similar source classes. We denote their corresponding learned tokens as $\{e_{c_1}, ..., e_{c_n}\}$, where each $e_{c_i} \in \mathbb{R}^D$. We begin by stacking these tokens into a single matrix $\theta_s \in \mathbb{R}^{n \times D}$.

Our objective is to compute a new guided token $\theta_{CTA} \in \mathbb{R}^D$, that captures the most characteristics from $\theta_s$. To achieve this, we employ an attention mechanism where the query is derived from the collective context of similar classes. Specifically, we compute the mean characteristic information $\overline{\theta}_s = \frac{1}{n} \sum_{j=1}^{n} \theta_s^j$, which encapsulates the "gist" of the semantic neighborhood. This combined information then attends to individual learned tokens in $\theta_s$ to generate attention scores. The final

token comprises the weighted information of all the learned similar classes' tokens.

$$\theta_{CTA} = softmax\left(\frac{\left(\frac{1}{n}\sum_{j=1}^{n}\theta_s^j\right) \times (\theta_s)^{\mathbf{T}}}{\sqrt{n}}\right)\theta_s \qquad (4)$$

where the scaling factor $\sqrt{n}$ ensures stable gradients. The softmax term produces a vector of attention weights, assigning importance to each source class and thus enabling selective knowledge transfer.

### 3.4.1 ENFORCING REPRESENTATIONAL SEPARATION VIA PERTURBATION

While Eqn. 4 provides a strong semantic prior, it poses a challenge: the initial guided token $\theta_{CTA}$ may lie too close to its source in the latent space. As the model trains on new data, this proximity can cause the latent space of the new class to overlap with that of its progenitors, leading to catastrophic forgetting and the misclassification of old classes as new ones during inference (illustrated in Tab. 3).

To mitigate this, we introduce a controlled perturbation to the synthesized token. Instead of directly using $\theta_{CTA}$, we regularize it by interpolating with a noisy version of itself, thereby enforcing a "buffer zone" in the latent space while maintaining semantic alignment. The final token is computed as:

$$\hat{\theta}_{CTA} = \alpha\theta_{CTA} + (1-\alpha)\mathcal{N}(\theta_{CTA}, \sigma^2) \qquad (5)$$

Here, $\mathcal{N}(\theta_{CTA}, \sigma^2)$ is a sample from a Gaussian distribution with mean $\theta_{CTA}$ and variance $\sigma^2$. The hyperparameter $\alpha \in [0,1]$ controls the trade-off between preserving the precise transferred knowledge (alignment) and introducing diversity (separation). Ablation studies on the impact of $\alpha$ and $\sigma$ are provided in §E. The complete training strategy for initialization and knowledge transfer is shown in §A.

### 3.5 OBJECTIVE FUNCTION

Our training objective for each individual step builds upon the strategy in Park et al. (2024). We employ a standard cross-entropy loss ($\mathcal{L}_{ce}$) for learning new classes and use feature ($\mathcal{L}_{fd}$) and knowledge ($\mathcal{L}_{kd}$) distillation losses to prevent general catastrophic forgetting.

However, our selective initialization brings the new class tokens into close proximity with their source classes, creating a risk of class confusion. To counteract this, we introduce a *Context Transfer Loss* ($\mathcal{L}_{ct}$), which enforces a margin-based separation. Given the new class token $e_{c_{new}}$ and the set of learned source class tokens $\{e_c\}$, where $c \in \mathcal{C}_s$, the loss is:

$$\mathcal{L}_{ct} = \frac{1}{|\mathcal{C}_s|}\sum_{c\in\mathcal{C}_s}\max(0, M - \|e_{c_{new}} - e_c\|_2) \qquad (6)$$

Here, $\|\cdot\|_2$ is the Euclidean distance between the new class token and a source class token. The loss only becomes positive if this distance is smaller than the predefined margin $M$, actively pushing them apart in latent space, as shown in our analysis in Fig. 1[Bottom]. The final objective is a weighted sum of these components: $\mathcal{L}_{total} = \mathcal{L}_{ce} + \lambda_{kd}\mathcal{L}_{kd} + \lambda_{fd}\mathcal{L}_{fd} + \lambda_{ct}\mathcal{L}_{ct}$. A detailed analysis of each component is provided in ablation study and §B.

## 4 EXPERIMENTS

### 4.1 EXPERIMENTAL SETUP

**Training setting.** We evaluate our proposed approach under two distinct experimental scenarios– *disjoint* and *overlapped* Park et al. (2024); Zhang & Gao (2024). In both scenarios, ground-truth labels are provided exclusively for the classes designated for the current task, $t$. The primary distinction lies in the composition of the input images: In the standard disjoint setting, images within the current task's dataset, $\mathcal{D}_t$, contain instances only from the current class set, $\mathcal{C}_{1:t}$; Conversely, the overlapped setting permits images to also contain unlabeled instances of objects from future classes ($\mathcal{C}_{1:T}$). Hence, the overlapped scenario more closely aligns with real-world data streams, presenting a significantly more challenging and realistic benchmark.

**Datasets and Evaluation Metrics.** Following prior works Wang et al. (2024); Kwak et al. (2024); Zhang & Gao (2024); Park et al. (2024), we conduct our experiments on two public benchmarks: Pascal VOC 2012 Everingham et al. (2015) and ADE20K Zhou et al. (2017). Pascal VOC comprises $10,582$ training and $1449$ testing images across 20 foreground categories. ADE20K is a larger-scale dataset with 150 classes, containing $20,210$ training and $2000$ testing images. For the Pascal VOC, we evaluate on the $15-1$, $15-5$ and $19-1$ splits under both disjoint and overlapped scenarios. The $15-1$ split, for example, consists of a base task with 15 classes, followed by 5 incremental tasks each adding a single new class, making a total of 6 tasks. For ADE20K, we use the $100-50$, $50-50$, $100-10$, and $100-5$ splits, focusing on the more realistic overlapped scenario. We measure the performance using mean intersection-over-union (mIoU) for the base classes, incremental classes, and all classes combined.

**Implementation Details.** Our approach is built upon a Vision Transformer (ViT-B/16) backbone Dosovitskiy et al. (2020), pre-trained on ImageNet. The segmentation head is a transformer-based decoder inspired by the Segmenter Strudel et al. (2021). Our training protocol largely follows the configuration in Park et al. (2024). We use the SGD optimiser and employ standard data augmentations. All experiments are implemented in PyTorch and conducted on a single NVIDIA A100 GPU. For Pascal VOC, we set the initial learning rate to 1e-3 and train for 32 epochs with a batch size of 16. For ADE20K, we use a learning rate of 5e-4 and train for 64 epochs with a batch size of 8. We set the model parameters $\alpha = 0.9$ and $\sigma = 0.05$. The threshold value $\xi$ is fixed at 0.15. For our proposed loss $\mathcal{L}_{ct}$, we set the margin M at 1.0, and its corresponding weight to 0.8. The weights for the other loss terms, $\mathcal{L}_{fd}$ and $\mathcal{L}_{kd}$, are adopted from Park et al. (2024)

## 4.2 EXPERIMENTAL RESULTS

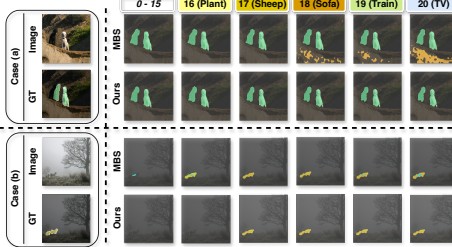

Figure 5: Visual comparison on $15-1$ scenario of the Pascal VOC between MBS and ours.

Experimental results in Table 2 illustrate the evaluation of our proposed approach on *Pascal VOC* and *ADE20K* under different training settings. We observe that our proposed approach outperforms almost all the previous works Zhang & Gao (2024); Shang et al. (2023); Cermelli et al. (2020); Cha et al. (2021); Xie et al. (2024); Park et al. (2024) across almost all settings by a significant margin. Particularly when compared with Xie et al. (2024), which focuses on previous knowledge as well, our method surpasses it across all the settings. Interestingly, in the *15-5* setting where multiple classes are added incrementally, learning in this setting is comparatively challenging compared to the rest, as multiple classes with varied distribution and number of images are included. Our approach, in this setting, surpasses previous works by a significant margin. We re-implement the results of Park et al. (2024) on our machine multiple times and provide their average results (denoted by ‡) for fair comparison. It is important to note that these results are different from the ones posted at Park et al. (2024). Primary qualitative results on Pascal are visualized in Fig. 5. Additional analyses with other challenging settings are provided in §D).

ADE20K is a significantly challenging dataset with more realistic and diverse settings. Hence, the results posted in this setting are substantially different from Pascal VOC. It is interesting to note that in our method, the margins between fundamental and incremental classes across all settings are significantly less than Xie et al. (2024). This shows that our approach to identifying the previous similar classes proves to be more effective than NeST and further maintains the trade-off between stability and plasticity. As for Park et al. (2024), our approach surpasses the re-implemented results by an average of 5.02% across all settings.

## 4.3 ABLATION STUDIES

In this section, we perform ablation studies under diverse settings to analyze the effectiveness of our proposed approach. We perform this study on the Pascal VOC dataset in the overlapped scenario for varied settings. Additional analyses are provided in the § E.

Table 2: Performance comparison under different scenarios for *overlapped* setting. ‡ implies results are reproduced from the official repository. † indicates results are excerpted from Park et al. (2024); Shang et al. (2023). Best two results are marked in ■ and ■ respectively. ■ represents the standard deviation after performing multiple runs.

| Methods | Backbone | Pascal VOC | | | | | | | | | ADE20K | | | | | | | | | | | | |
|---|---|---|---|---|---|---|---|---|---|---|---|---|---|---|---|---|---|---|---|---|---|---|---|
| | | 19-1 (2 tasks) | | | 15-5 (2 tasks) | | | 15-1 (6 tasks) | | | 100-50 (2 tasks) | | | 50-50 (3 tasks) | | | 100-10 (6 tasks) | | | 100-5 (11 tasks) | | |
| | | 1-19 | 20 | All | 1-15 | 16-20 | All | 1-15 | 16-20 | All | 1-100 | 101-150 | All | 1-50 | 51-150 | All | 1-100 | 101-150 | All | 1-100 | 101-150 | All |
| **CNN-based Network** | | | | | | | | | | | | | | | | | | | | | | |
| MiB† Cermelli et al. (2020) | Res101 | 70.2 | 22.1 | 67.8 | 75.5 | 49.4 | 69.0 | 35.1 | 13.5 | 29.7 | 40.5 | 17.2 | 32.8 | 45.5 | 21.0 | 29.3 | 38.2 | 11.1 | 29.2 | 36.0 | 5.7 | 26.0 |
| SSUL Cha et al. (2021) | Res101 | 77.8 | 49.8 | 76.5 | 78.4 | 55.8 | 73.0 | 78.4 | 49.0 | 71.4 | 42.8 | 17.5 | 34.5 | 49.1 | 20.1 | 29.8 | 42.9 | 17.7 | 34.5 | 42.9 | 17.8 | 34.6 |
| SPPA Lin et al. (2022) | Res101 | 76.5 | 36.2 | 74.6 | 78.1 | 52.9 | 72.1 | 66.2 | 23.3 | 56.0 | 42.9 | 19.9 | 35.2 | 49.8 | 23.9 | 32.5 | 41.0 | 12.5 | 31.5 | - | - | - |
| RCIL† Zhang et al. (2022b) | Res101 | 77.0 | 31.5 | 74.7 | 78.8 | 52.0 | 72.4 | 70.6 | 23.7 | 59.4 | 42.3 | 18.8 | 34.5 | 48.3 | 25.0 | 32.5 | 39.3 | 17.6 | 32.0 | 38.5 | 11.5 | 29.6 |
| IDEC Zhao et al. (2023) | Res101 | - | - | - | 78.0 | 51.8 | 71.8 | 77.0 | 36.5 | 67.3 | 42.0 | 18.2 | 34.1 | 47.4 | 26.0 | 33.1 | 40.3 | 17.6 | 32.7 | 39.2 | 14.6 | 31.0 |
| LGKD+PLOP Yang et al. (2023) | Res101 | 76.5 | 42.9 | 75.7 | 78.7 | 56.1 | 73.9 | 69.3 | 30.9 | 61.1 | 43.6 | 25.7 | 37.5 | 49.4 | 29.4 | 36.0 | 42.1 | 22.0 | 35.4 | - | - | - |
| STAR Chen et al. (2023b) | Res101 | 78.0 | 47.1 | 76.5 | 79.5 | 58.9 | 74.6 | 79.5 | 50.6 | 72.6 | 42.4 | 24.2 | 36.4 | 48.7 | 27.2 | 34.4 | 42.0 | 20.6 | 34.9 | - | - | - |
| REMINDER Phan et al. (2022) | Res101 | 76.5 | 32.3 | 74.4 | 76.1 | 50.7 | 70.1 | 68.3 | 27.2 | 58.5 | 41.6 | 19.2 | 34.1 | 47.1 | 20.4 | 29.4 | 39.0 | 21.3 | 33.1 | 36.1 | 16.4 | 29.5 |
| PLOP + NeST Xie et al. (2024) | Res101 | 77.0 | 49.1 | 75.7 | 77.6 | 55.8 | 72.4 | 72.2 | 33.7 | 63.1 | 42.2 | 24.3 | 36.3 | 48.7 | 27.7 | 34.8 | 40.9 | 22.0 | 34.7 | 39.3 | 17.4 | 32.0 |
| RCIL + NeST Xie et al. (2024) | Res101 | 77.0 | 33.3 | 74.9 | 79.0 | 52.8 | 72.8 | 71.9 | 28.0 | 61.4 | 42.3 | 22.8 | 35.8 | 48.2 | 27.4 | 34.4 | 40.7 | 19.0 | 33.5 | 39.4 | 15.5 | 31.5 |
| BARM Zhang & Gao (2024) | Res101 | 78.2 | 42.2 | 76.4 | - | - | - | 77.6 | 45.9 | 70.0 | 42.0 | 23.0 | 35.7 | 47.9 | 26.5 | 33.7 | 41.1 | 23.1 | 35.2 | 40.5 | 21.2 | 34.1 |
| ADAPTER Zhu et al. (2024) | Res101 | 78.0 | 50.7 | 76.7 | 79.7 | 59.7 | 75.0 | 79.9 | 51.9 | 73.2 | 43.1 | 23.6 | 36.7 | 49.3 | 27.3 | 34.7 | 42.9 | 19.9 | 35.3 | 42.6 | 18.0 | 34.5 |
| **Transformer-based Network** | | | | | | | | | | | | | | | | | | | | | | |
| MiB† Cermelli et al. (2020) | ViT | 79.9 | 47.7 | 79.1 | 78.6 | 63.1 | 75.6 | 72.6 | 23.1 | 61.7 | 46.4 | 35.0 | 42.6 | 52.2 | 35.6 | 41.1 | 43.0 | 30.8 | 38.9 | 40.2 | 26.6 | 35.7 |
| CoinSeg Zhang et al. (2023) | Swin-B | 81.5 | 44.8 | 79.8 | 82.1 | 63.2 | 77.6 | 82.7 | 52.5 | 75.5 | 41.6 | 26.7 | 36.6 | 49.0 | 28.9 | 35.6 | 42.1 | 24.5 | 36.2 | 43.1 | 24.1 | 36.8 |
| INC† Shang et al. (2023) | ViT | 82.5 | 61.0 | 82.1 | 82.5 | 69.2 | 79.9 | 79.6 | 59.6 | 75.6 | 49.4 | 35.6 | 44.8 | 56.2 | 37.8 | 43.9 | 48.5 | 34.6 | 43.9 | 46.9 | 31.3 | 41.7 |
| MiB + NeST Xie et al. (2024) | Swin-B | 79.7 | 60.0 | 78.8 | 81.2 | 67.4 | 77.9 | 77.0 | 53.3 | 71.4 | 42.8 | 27.8 | 37.9 | 49.7 | 29.3 | 36.2 | 41.8 | 23.8 | 35.9 | 40.5 | 19.9 | 33.7 |
| PLOP + NeST Xie et al. (2024) | Swin-B | 79.6 | 70.2 | 79.1 | 80.5 | 70.8 | 78.2 | 76.8 | 57.2 | 72.2 | 43.5 | 26.5 | 37.9 | 50.6 | 28.9 | 36.2 | 41.7 | 24.2 | 35.9 | 39.7 | 18.3 | 32.6 |
| MBS‡ Park et al. (2024) | ViT | 81.5 | 67.0 | 80.8 | 82.7 | 74.0 | 80.5 | 82.3 | 69.0 | 79.0 | 47.7 | 35.6 | 43.7 | 54.4 | 37.1 | 42.9 | 47.7 | 31.5 | 42.3 | 44.4 | 22.1 | 38.8 |
| | | ±0.2 | ±0.4 | ±0.4 | ±0.7 | ±0.7 | ±0.6 | ±0.3 | ±0.8 | ±0.5 | ±0.8 | ±0.2 | ±0.6 | ±0.8 | ±0.1 | ±0.8 | ±0.7 | ±0.4 | ±0.2 | ±0.1 | ±0.2 | ±0.3 |
| Ours | | 81.6 | 68.2 | 80.9 | 83.9 | 76.0 | 81.6 | 83.3 | 72.0 | 80.5 | 49.1 | 36.8 | 45.0 | 56.4 | 38.7 | 44.6 | 48.0 | 34.8 | 43.9 | 46.7 | 27.5 | 41.1 |

Table 3: Ablation study on Pascal VOC, $15-1$ setting. $\theta_{best}$: token of the most similar class, $\theta_{Avg}$: average token of all similar classes, and $\theta_{CTA}$: adaptive token using Context Transfer Attention. **bold** represents the best results. Highlighted columns show the performance drops that were part of a similar class set for a new class.

| Methods | 1-15 | | | | | | | | | | | | | | | | 16-20 | | | | | |
|---|---|---|---|---|---|---|---|---|---|---|---|---|---|---|---|---|---|---|---|---|---|---|
| | aero | bike | bird | boat | bottle | bus | car | cat | chair | cow | table | dog | horse | motor | person | Avg | plant | sheep | sofa | train | tv | Avg |
| $\theta_{best}$ | 0.0 | 44.7 | 92.6 | 19.7 | 87.1 | 91.5 | 87.4 | 93.0 | 46.2 | 53.3 | 57.3 | 90.3 | 91.1 | 91.2 | 89.0 | 69.0 | 39.7 | 55.8 | 40.1 | 55.3 | 57.6 | 49.7 |
| $\theta_{Avg}$ | 0.0 | 44.3 | 91.9 | 76.1 | 86.3 | 93.6 | 87.9 | 95.5 | 50.4 | 71.5 | 62.3 | 92.3 | 90.0 | 89.4 | 88.9 | 74.7 | 56.1 | 0.1 | 36.6 | 43.4 | 62.6 | 41.3 |
| $\theta_{Avg}$ w $\mathcal{N}$ | 89.6 | 45.3 | 93.1 | 77.1 | 85.4 | 90.9 | 87.6 | 95.3 | 45.9 | 4.4 | 56.7 | 92.5 | 84.6 | 89.2 | 88.6 | 75.1 | 70.0 | 0.0 | 30.9 | 44.5 | 59.6 | 41.0 |
| $\theta_{CTA}$ w/o $\mathcal{N}$ | 16.0 | 44.7 | 87.6 | 80.1 | 78.9 | 92.4 | 88.5 | 94.4 | 50.2 | 48.5 | 63.2 | 92.5 | 90.7 | 89.3 | 88.7 | 73.7 | 72.1 | 9.3 | 44.5 | 42.1 | 57.7 | 45.1 |
| $\theta_{CTA}$ (Ours) | 93.7 | 49.1 | 90.3 | 81.4 | 87.8 | 95.2 | 92.0 | 95.3 | 51.9 | 90.9 | 64.1 | 94.2 | 91.3 | 90.7 | 89.4 | **83.9** | 68.2 | 89.0 | 49.6 | 86.9 | 67.3 | **72.0** |

**Effectiveness of knowledge transfer.** In Table 3, we evaluate the class-wise performance to analyze the most efficient approach for knowledge transfer. Specifically, we investigate how transferring knowledge from previously learned similar classes affects the performance of both similar and dissimilar past classes. To this end, we compare our proposed attention-based approach $\theta_{CTA}$ to two major baselines (i) $\theta_{Best}$ (selects the most similar class from the subset) and (ii) $\theta_{Avg}$ (averages the representations of all similar classes). Additionally, we also evaluate the impact of integrating controlled noise $\mathcal{N}$ across the strategies. Primarily, we observe that our proposed approach consistently outperforms the baselines by a significant margin. In particular, our attention-based framework achieves at least 11% and 41% improvement over the baselines on the $1-15$ and $16-20$ tasks, respectively.

Upon looking closely, in $\theta_{Best}$ and $\theta_{Avg}$ methods, transferring knowledge from certain classes (*e.g.* aero) leads to a substantial drop in its performance–highlighted in ■. Several other classes also exhibit similar degradation. Our proposed approach ensures that previous similar classes retain their knowledge while also adapting to the new ones effectively by accommodating a controlled noise component. Furthermore, it is also evident that adding the noise component improves the overall performance across all classes.

**Effectiveness of loss functions.** To quantify the contribution of each loss component ($\mathcal{L}_{ce}$, $\mathcal{L}_{fd}$, $\mathcal{L}_{kd}$, $\mathcal{L}_{ct}$) within our CISS framework, we conduct a detailed ablation study across $15-1$ and $15-5$ scenarios, with results presented in Table 4. Removing the feature distillation loss ($\mathcal{L}_{fd}$) or the knowledge distillation loss ($\mathcal{L}_{kd}$) leads to a catastrophic decline in performance, particularly on incremental classes. In the 15-1 setting, ablating $\mathcal{L}_{fd}$ causes the mIoU on new classes to plummet from 72.0% to 14.2%. Similarly, removing $\mathcal{L}_{kd}$ drops the same metric to 32.6%. This underscores that $\mathcal{L}_{fd}$ is vital for learning high-quality, patch-level representations of novel classes, while $\mathcal{L}_{kd}$ is indispensable for mitigating catastrophic forgetting of the base classes. Our proposed contrastive loss provides a substantial performance boost. Removing $\mathcal{L}_{ct}$ degrades

Table 4: Ablation study on the Pascal VOC dataset about the effectiveness of each loss function. **Bold** denotes the best result.

| $\mathcal{L}_{ce}$ | $\mathcal{L}_{fd}$ | $\mathcal{L}_{kd}$ | $\mathcal{L}_{ct}$ | 15-1 (6 tasks) | | | 15-5 (2 tasks) | | |
|---|---|---|---|---|---|---|---|---|---|
| | | | | 1-15 | 16-20 | All | 1-15 | 16-20 | All |
| ✓ | ✗ | ✓ | ✓ | 48.5 | 14.2 | 52.2 | 83.0 | 70.9 | 80.6 |
| ✓ | ✓ | ✗ | ✓ | 63.0 | 32.6 | 60.0 | 74.3 | 57.2 | 71.1 |
| ✓ | ✓ | ✓ | ✗ | 80.1 | 60.7 | 76.1 | 83.2 | 70.2 | 80.0 |
| ✓ | ✗ | ✗ | ✗ | 57.1 | 8.7 | 52.2 | 77.1 | 60.1 | 73.8 |
| ✓ | ✗ | ✗ | ✓ | 53.2 | 10.5 | 55.4 | 75.4 | 58.7 | 72.3 |
| ✓ | ✓ | ✗ | ✗ | 61.9 | 17.5 | 61.6 | 72.2 | 53.7 | 68.8 |
| ✓ | ✗ | ✓ | ✗ | 79.3 | 24.4 | 66.8 | **84.1** | 73.7 | **82.1** |
| ✓ | ✓ | ✓ | ✓ | **83.3** | **72.0** | **80.5** | 83.9 | **76.0** | 81.6 |

the overall mIoU from 80.5% to 76.1% in the 15-1 setting, with the most significant damage inflicted on the incremental classes (72.0% $\rightarrow$ 60.7%).

## CONCLUSION

In this work, we introduced **SELECT**, a novel framework that addresses an adaptive knowledge transfer strategy. By identifying semantic similarities between old and new classes and employing a selective transfer mechanism, SELECT provides a more principled approach to knowledge utilization. Our proposed loss function further strengthens the model by encouraging distinctiveness between transferred knowledge and the original representations of influential old classes. Finally, our empirical results demonstrate the effectiveness of our proposed approach.

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

# A COMPLETE TRAINING PIPELINE

---

**Algorithm 1** Training Strategy for the proposed SELECT.

---

**Require:** Training dataset $\mathcal{D}_t = \{(x_i, y_i)\}_t$ for current task $t$, model with previous task's weights $f_{t-1}$, current task's model $f_t$, Total tasks $T$.
**Ensure:** Initial class representation for new task $\theta_{CTA}$
1: **for** $t \in \{1, 2, \ldots, T\}$ **do**
2:     **for** $(x_i, y_i) \in \mathcal{D}_t$ **do**                       ▷ Class similarity detection. §3.3
3:         $\mathcal{C}_s \leftarrow ClassSimilarity(x_i, y_i, f_{t-1})$
4:     **end for**
5:     $\theta_s \leftarrow \mathcal{C}_s$                    ▷ Corresponding tokens of previous similar classes.
6:     $\hat{\theta}_{CTA} \leftarrow Attention(\theta_s)$                ▷ Context transfer attention. §3.4
7:     $f_t \leftarrow Update(f_t | \hat{\theta}_{CTA})$
8:     **while** not converged **do**                ▷ Training incremental setting.
9:         train $f_t$
10:     **end while**
11: **end for**

---

# B LOSS FUNCTION

In this section, we elaborate on the loss functions Park et al. (2024) discussed in §3.5.

**Cross-Entropy Loss ($\mathcal{L}_{ce}$):** The cross-entropy loss is used in both base and incremental tasks. In the base task ($t = 0$), we use the conventional cross-entropy loss between the predicted mask $\hat{y}_0 \in \mathbb{R}^{H \times W \times (0:\mathcal{C}_t)}$ and its corresponding ground truth $y_t \in \mathbb{R}^{H \times W}$:

$$\mathcal{L}_{ce}(y_0, \hat{y}_0) = \frac{1}{HW} \sum_{i=1}^{HW} y_{0,i} \log \hat{y}_{0,i} \tag{7}$$

where $H$ and $W$ are the height and width of an image. In the incremental task ($t > 0$), we use the previous task's predicted label along with the ground truth of the current class as the pseudo label $\tilde{y}_t(y_t, \hat{y}_{t-1})$ and calculate the loss with the combined predicted label $\hat{y}_t$. The updated $\mathcal{L}_{ce}$ is:

$$\mathcal{L}_{ce}(\tilde{y}_t, \hat{y}_t) = \frac{1}{HW} \sum_{i=1}^{HW} \tilde{y}_{t,i}(y_t, \hat{y}_{t-1}) \log \hat{y}_{t,i} \tag{8}$$

**Feature Distillation Loss ($\mathcal{L}_{fd}$):** We employ this loss (in incremental task) to prevent the current model's features from deviating from the previous ones. This loss is calculated between the output feature patches from the decoder of both the previous $z_{dec,t-1}$ and current models $z_{dec,t}$.

$$\mathcal{L}_{fd} = \frac{1}{HW} \sum_{i=1}^{HW} \|z_{dec,t-1,i} - z_{dec,t,i}\|^2 \tag{9}$$

**Knowledge Distillation Loss ($\mathcal{L}_{kd}$):** We employ this loss (in incremental task) to distil the previous model's prediction and transfer the background predictions to the new class. This loss is calculated between the output predictions of both the previous $\hat{y}_{t-1}$ and current models $\hat{y}_t$.

$$\mathcal{L}_{kd} = -\frac{1}{HW} \sum_{i=1}^{HW} \hat{y}_{t-1,i} \log \hat{y}_{t,i} \tag{10}$$

The final objective function for our proposed work is:

$$\mathcal{L}_{total} = \mathcal{L}_{ce} + \lambda_{fd} * \mathcal{L}_{fd} + \lambda_{kd} * \mathcal{L}_{kd} + \lambda_{ct} * \mathcal{L}_{ct} \tag{11}$$

where $\lambda_{fd}$, $\lambda_{kd}$, and $\lambda_{ct}$ are the respective hyperparameters for $\mathcal{L}_{fd}$, $\mathcal{L}_{kd}$, and $\mathcal{L}_{ct}$.

## C ADDITIONAL ANALYSIS ON CLASS SIMILARITY DETECTION

In §3.3, we explained the strategy for identifying similar classes from the previously learned classes for the current task $t$. In this section, we will further analyze the effectiveness of utilizing similar classes and their impact on the overall performance.

### C.1 IMAGE-WISE DESPARITY

In Fig. 4, we visualized the strong semantic relevance and lower average perturbation across some classes in the new dataset. In Fig. 6, we randomly sample a few images (from current task's dataset) and visualize these perturbations.

It is evident that all the images have the deviation across the same classes, where some classes have minimal deviation, showing large similarity in their characteristics. This similarity is utilized to identify the most similar class.

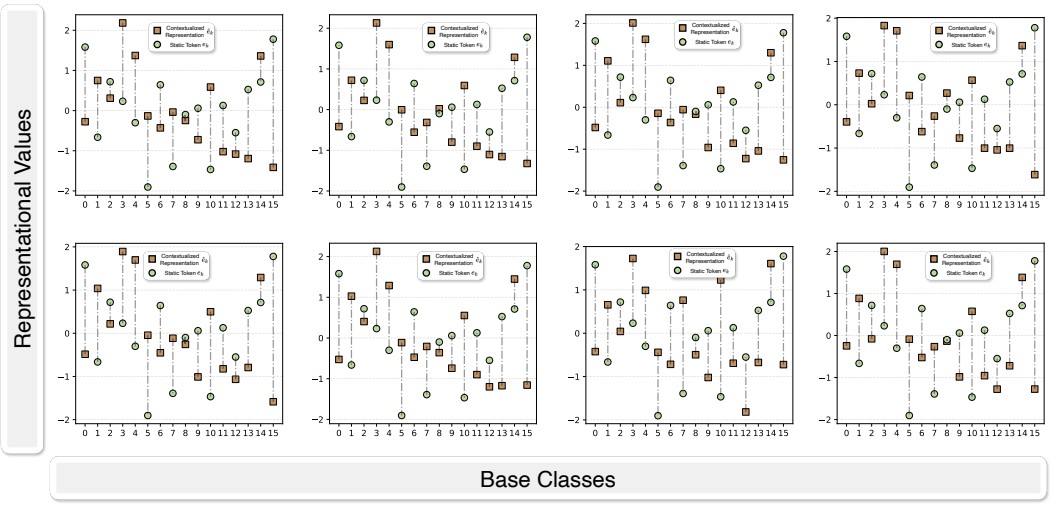

Figure 6: Visualization of representational perturbation between static token $e_k$ and contextualized representation $\hat{e}_k$ for randomly sampled images.

### C.2 STATISTICAL ANALYSIS ON SEMANTIC SIMILARITY

Our proposed approach leverages the characteristics from a well-trained representational space where features are not random but semantically clustered.

When we compute the interaction between an incremental class image $x_i$ (*e.g.* "Sheep") and a base class static token $e_k$ (*e.g.* "Dog"). The "sheep" image contains visual features (snout, four legs) that are compatible with the Dog token. The decoder finds strong alignment, resulting in a refined token $\hat{e}_k^{x_i}$ that remains close to the original $e_k$ in the representational space. The deviation (Euclidean distance) is therefore small.

If we pair the same "Sheep" image with a dissimilar token $e_k$ (*e.g.* "Car"), the image features actively conflict with the token's expected features (metal, wheels). This context mismatch forces the resulting representation $\hat{e}_k^{x_i}$ to drift significantly from $e_k$. This deviation is therefore large.

To validate this claim, we generated a class similarity correlation heatmap (Fig. 7). The cell color represents the calculated similarity metric (inverse of deviation) between the Base and the new incremental classes.

The heatmap reveals distinct, semantically coherent clusters rather than random noise. Particularly, the incremental class "Sheep" shows a high similarity score with base classes like "Cow", "Horse", "Dog", and "Cat", but a near-zero score with "Aero" or "Motor".

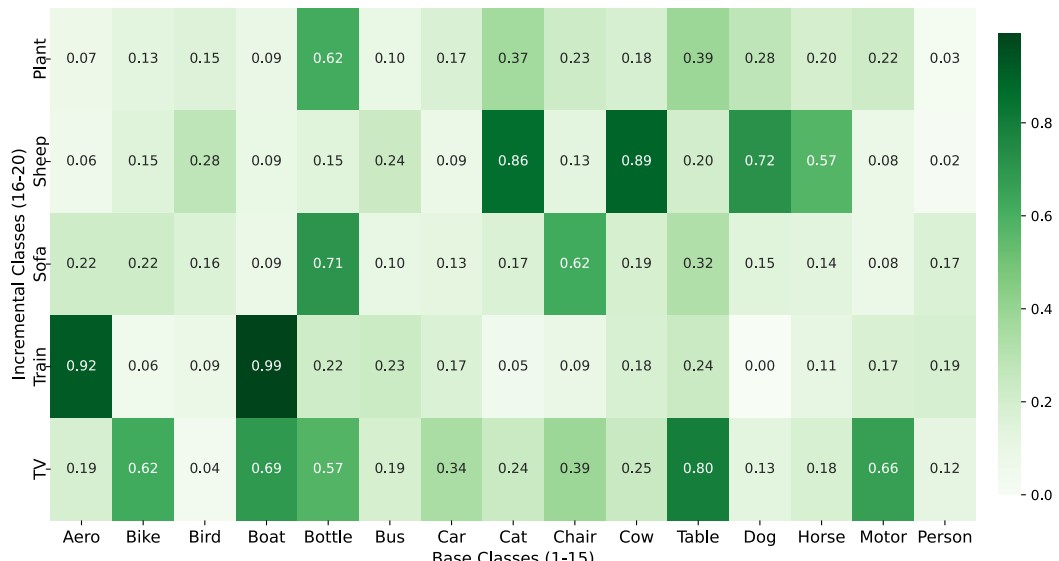

Figure 7: Visual analysis of class similarity using their semantic characteristics.

## C.3 SIMILAR CLASS DISTRIBUTION

In Fig. 8, we provide the visualization of class distribution for all incremental scenarios in the $15 - 1$ setting. From the figure, it is evident that there is a selective number of classes in each incremental class, which resembles the latter. Furthermore, we also provide an ablation study that compares the performance between similar and dissimilar classes (in Table 9).

Table 5: Additional performance comparison on Pascal VOC under different scenarios for *overlapped* setting. ‡ implies results are reproduced from the official repository. † indicates results are excerpted from Park et al. (2024); Shang et al. (2023). Best two results are marked in ■ and ■ respectively.

| Methods | 19-1 (2 tasks) | | | 15-5 (2 tasks) | | | 15-1 (6 tasks) | | | 10-1 (11 tasks) | | | 5-3 (6 tasks) | | |
|---|---|---|---|---|---|---|---|---|---|---|---|---|---|---|---|
| | 1-19 | 20 | All | 1-15 | 16-20 | All | 1-15 | 16-20 | All | 1-10 | 11-20 | All | 1-5 | 6-20 | All |
| **CNN-based Network** | | | | | | | | | | | | | | | |
| MiB[†] Cermelli et al. (2020) | 70.2 | 22.1 | 67.8 | 75.5 | 49.4 | 69.0 | 35.1 | 13.5 | 29.7 | 20.0 | 20.1 | 20.1 | 57.1 | 42.6 | 46.7 |
| PLOP[†] Douillard et al. (2021) | 75.4 | 37.4 | 73.5 | 75.7 | 51.7 | 70.1 | 65.1 | 21.1 | 54.6 | 44.0 | 15.5 | 30.5 | 17.5 | 19.2 | 18.7 |
| RCIL[†] Zhang et al. (2022b) | 77.0 | 31.5 | 74.7 | 78.8 | 52.0 | 72.4 | 70.6 | 23.7 | 59.4 | 55.4 | 15.1 | 34.3 | - | - | - |
| SSUL Cha et al. (2021) | 77.8 | 49.8 | 76.5 | 78.4 | 55.8 | 73.0 | 78.4 | 49.0 | 71.4 | 74.0 | 53.2 | 64.1 | 71.3 | 53.2 | 58.4 |
| PLOP + Cs²K Cong et al. (2024) | - | - | - | - | - | - | 77.9 | 46.4 | 70.4 | 74.4 | 47.2 | 61.5 | 58.4 | 53.4 | 54.8 |
| MiB + NeST Xie et al. (2024) | 71.7 | 28.2 | 69.7 | 77.1 | 50.1 | 70.7 | 61.7 | 20.4 | 51.8 | 52.3 | 21.0 | 37.4 | - | - | - |
| PLOP + NeST Xie et al. (2024) | 77.0 | 49.1 | 75.7 | 77.6 | 55.8 | 72.4 | 72.2 | 33.7 | 63.1 | 54.2 | 17.8 | 36.9 | - | - | - |
| RCIL + NeST Xie et al. (2024) | 77.0 | 33.3 | 74.9 | 79.0 | 52.8 | 72.8 | 71.9 | 28.0 | 61.4 | 51.4 | 20.9 | 36.8 | - | - | - |
| BARM Zhang & Gao (2024) | 78.2 | 42.2 | 76.4 | - | - | - | 77.6 | 45.9 | 70.0 | 72.2 | 51.0 | 62.1 | 71.3 | 57.0 | 61.1 |
| IDEC Zhao et al. (2023) | - | - | - | 78.0 | 51.8 | 71.8 | 77.0 | 36.5 | 67.3 | 70.7 | 46.3 | 59.1 | 67.1 | 49.0 | 54.1 |
| STAR Chen et al. (2023b) | 78.0 | 47.1 | 76.5 | 79.5 | 58.9 | 74.6 | 79.5 | 50.6 | 72.6 | 73.1 | 55.4 | 64.7 | 71.9 | 61.5 | 64.4 |
| ADAPTER Zhu et al. (2024) | 78.0 | 50.7 | 76.7 | 79.7 | 59.7 | 75.0 | 79.9 | 51.9 | 73.2 | 74.9 | 54.3 | 65.1 | 73.8 | 61.9 | 65.3 |
| **Transformer-based Network** | | | | | | | | | | | | | | | |
| MiB[†] Cermelli et al. (2020) | 79.9 | 47.7 | 79.1 | 78.6 | 63.1 | 75.6 | 72.6 | 23.1 | 61.7 | - | - | - | 33.4 | 43.2 | 42.9 |
| RBC[†] Zhao et al. (2022) | 80.2 | 38.8 | 79.0 | 78.9 | 62.0 | 75.5 | 75.9 | 40.2 | 68.2 | - | - | - | - | - | - |
| INC[†] Shang et al. (2023) | 82.5 | 61.0 | 82.1 | 82.5 | 69.2 | 79.9 | 79.6 | 59.6 | 75.6 | 77.6 | 60.3 | 70.2 | - | - | - |
| MiB + NeST Xie et al. (2024) | 79.7 | 60.0 | 78.8 | 81.2 | 67.4 | 77.9 | 77.0 | 53.3 | 71.4 | 65.2 | 35.8 | 51.2 | - | - | - |
| PLOP + NeST Xie et al. (2024) | 79.6 | 70.2 | 79.1 | 80.5 | 70.8 | 78.2 | 76.8 | 57.2 | 72.2 | 64.3 | 28.3 | 47.2 | - | - | - |
| MBS[‡] Park et al. (2024) | 81.5 | 67.0 | 80.8 | 82.7 | 74.0 | 80.5 | 82.3 | 69.0 | 79.0 | 80.3 | 72.0 | 77.0 | 76.7 | 77.3 | 77.0 |
| Ours | 81.6 | 68.2 | 80.9 | 83.9 | 76.0 | 81.6 | 83.3 | 72.0 | 80.5 | 79.1 | 52.5 | 70.4 | 77.9 | 75.1 | 76.7 |

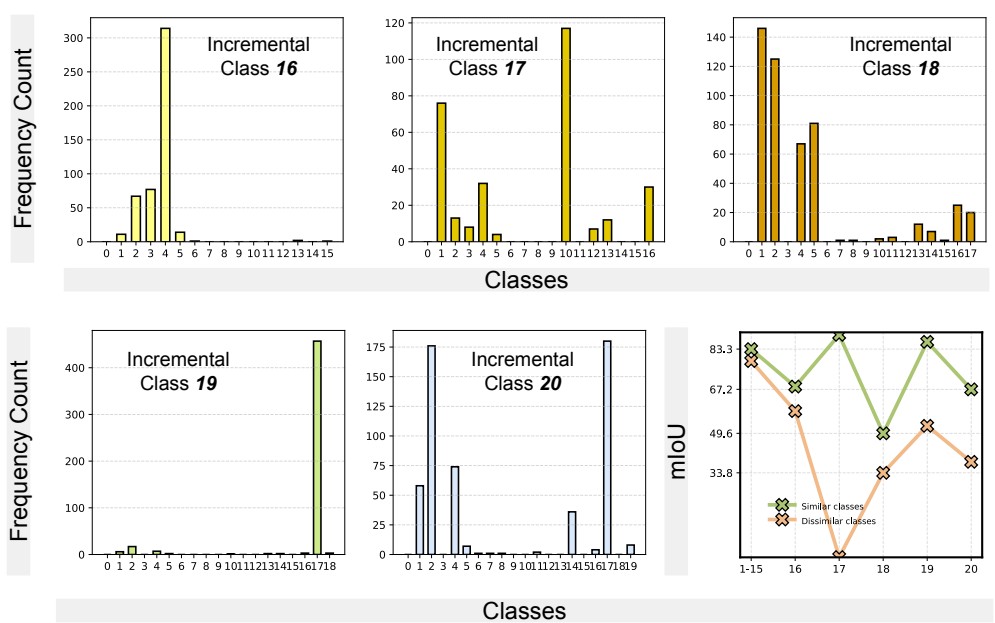

Figure 8: Visualization of class distribution across the incremental scenario in the $15 - 1$ setting. Additionally, there is a visual comparison of similar and dissimilar classes.

Table 6: Additional performance comparison on Pascal VOC under different scenarios for *disjoint* setting. ‡ implies results are reproduced from the official repository. † indicates results are excerpted from Park et al. (2024); Shang et al. (2023). Best two results are marked in ■ and ■ respectively.

| Methods | 19-1 (2 tasks) | | | 15-5 (2 tasks) | | | 15-1 (6 tasks) | | |
|---|---|---|---|---|---|---|---|---|---|
| | 1-19 | 20 | All | 1-15 | 16-20 | All | 1-15 | 16-20 | All |
| **CNN-based Network** | | | | | | | | | |
| MiB† Cermelli et al. (2020) | 69.6 | 25.6 | 67.4 | 71.8 | 43.3 | 64.7 | 46.2 | 12.9 | 37.9 |
| PLOP† Douillard et al. (2021) | 75.4 | 38.9 | 73.6 | 71.0 | 42.8 | 64.3 | 57.9 | 13.7 | 46.5 |
| RCIL† Zhang et al. (2022b) | - | - | - | 75.0 | 42.8 | 67.3 | 66.1 | 18.2 | 54.7 |
| SPPA Lin et al. (2022) | 75.5 | 38.0 | 73.7 | 75.3 | 48.7 | 69.0 | 59.6 | 15.6 | 49.1 |
| STAR Chen et al. (2023b) | 77.9 | 43.4 | 76.2 | 78.4 | 57.4 | 73.4 | 78.1 | 46.6 | 70.6 |
| ADAPTER Zhu et al. (2024) | 78.0 | 46.1 | 76.5 | 78.9 | 58.2 | 73.9 | 78.6 | 49.0 | 71.5 |
| **Transformer-based Network** | | | | | | | | | |
| MiB† Cermelli et al. (2020) | 80.6 | 45.2 | 79.6 | 75.0 | 59.9 | 72.3 | 66.7 | 26.3 | 58.3 |
| RBC† Zhao et al. (2022) | 80.9 | 42.1 | 79.7 | 77.7 | 59.1 | 74.0 | 69.0 | 28.4 | 60.5 |
| INC† Shang et al. (2023) | **82.4** | 64.2 | **82.2** | **81.6** | 62.2 | **77.6** | **81.4** | 57.1 | **76.2** |
| MBS‡ Park et al. (2024) | 81.5 | **65.0** | 80.7 | 80.5 | **64.1** | 76.4 | 79.6 | **57.8** | 74.1 |
| Ours | 81.9 | **66.1** | 81.1 | **82.2** | **67.9** | **78.6** | **80.7** | **61.9** | **75.7** |

# D  ADDITIONAL EXPERIMENTAL ANALYSIS

## D.1  DISCUSSION ON GAPS IN REPRODUCING THE RESULTS

In Table 2, 5, and 6, we reproduce the results of MBS for fair and direct comparison. As mentioned in § 4.2, we observe a discrepancy between the results reported in the original MBS paper and those we obtained from our reproduction. Below we provide the details of our reproduction process.

Our primary goal was to establish a fair and consistent experimental framework for all methods. We obtained the reproduced MBS results by running the official, publicly available source code provided

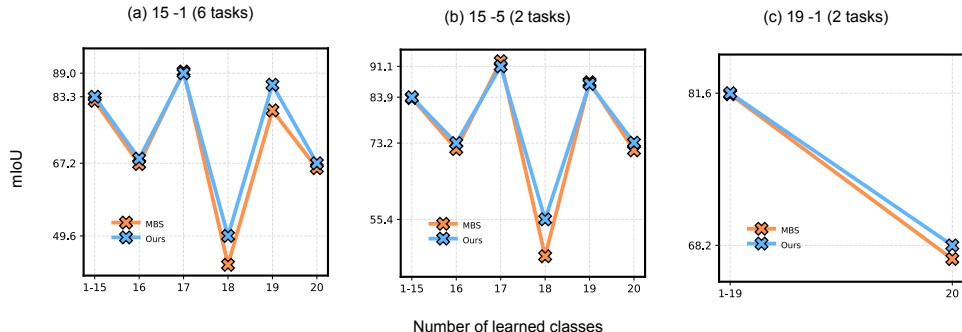

Figure 9: Incremental class-wise performance across different settings for Pascal VOC dataset.

by the MBS. We used their implementation details without modifying the core architecture or loss functions. To ensure a true comparison, our proposed method (SELECT) and baseline MBS were trained and evaluated within the same, unified environment. The reproduced scores are the average mIoU values obtained from multiple runs. The average variance observed for MBS was $\pm0.57$ on VOC and $\pm0.43$ on ADE20K (in Table 2).

## D.2 INDIVIDUAL INCREMENTAL CLASS ANALYSIS.

Fig. 9 visualizes the performance of individual incremental classes. We compare the performance of our proposed method with Park et al. (2024) (results are taken from Table 2. As shown in Fig. 9, we observe distinct behaviour across the settings in the Pascal benchmark. To start off, both methods achieve similar performance in the fundamental classes across all settings ("$1-15$" in $15-1$ or $15-5$ settings and "$1-19$" in $19-1$ setting). However, in the incremental scenario, we observe a significant performance gap between the two approaches. Specifically, we see a consistent performance drop in Park et al. (2024) across all settings. As seen in Fig. 9, new incremental classes adapt knowledge from previously learned similar classes. Perhaps this setting helps adapt to new classes in a more efficient manner.

Table 7: Additional performance comparison on Pascal VOC under transformer-based backbones for *overlapped* setting. Best results are marked in **Bold**.

| Models | Backbone | 15-1 | 15-5 | 10-1 |
|---|---|---|---|---|
| Incrementer | ViT-B | 75.5 | 79.9 | 70.2 |
| MiB | ViT-B | 53.5 | 80.2 | 25.5 |
| MiB + NeST | ViT-B | 76.5 | 80.3 | **71.9** |
| Ours | ViT-B | **80.5** | **81.6** | 70.4 |

## D.3 COMPARISONS WITH TRANSFORMER-BASED METHODS.

In Table 2, we originally reported the Swin-B results for NeST to align with the primary results highlighted in the original NeST paper. In Table 7, we conduct a fair evaluation with the ViT-B backbone as well. The results presented in the table are averaged across all the classes. These benchmark results are extracted from Xie et al. (2024). Our approach significantly outperformed NeST in almost all tasks.

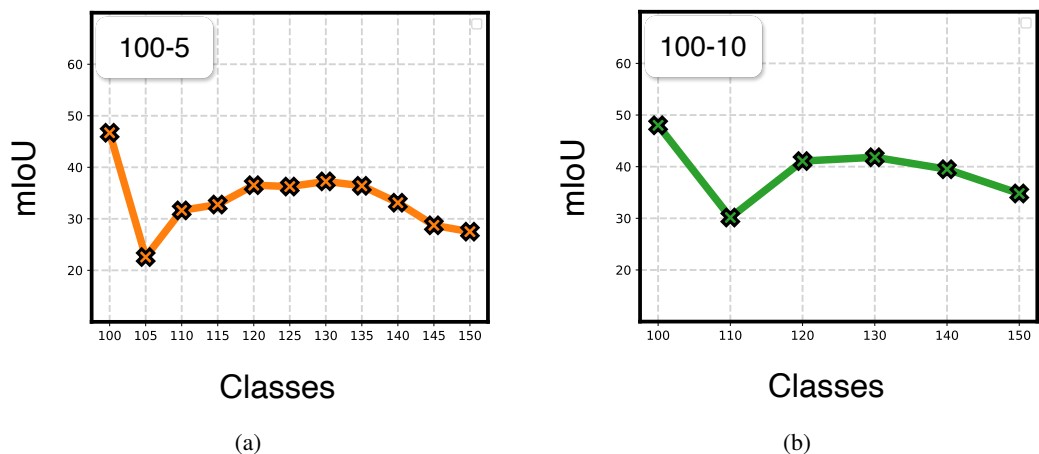

Figure 10: Analysis over longer task sequences (a) 100-5, and (b) 100-10 for the ADE20K dataset.

### D.4 OTHER CHALLENGING SETTINGS.

We perform additional experiments on the Pascal VOC dataset in the overlapped scenario ($10 - 1$ and $5 - 3$ settings), and disjoint scenario ($15 - 1$, $15 - 5$, and $19 - 1$ settings). Table 5 and Table 6 show our quantitative results for the same, respectively.

The $10 - 1$ and $5 - 3$ overlapped settings bring additional challenges to the class-incremental scenario. In both settings, the number of fundamental classes learned is less than that observed in the previous settings. Particularly in $5 - 3$, which has the least number of fundamental classes, there are multiple classes in a single incremental task, making learning from knowledge transfer comparatively difficult. Regarding the disjoint setting, our approach surpasses almost all previous works by a substantial margin.

### D.5 ANALYZING PROPOSED STRATEGY ON DIFFERENT ARCHITECTURES.

In this section, we provide an analysis to generalize our proposed strategy to CNN architectures. To validate this, we integrated our method into two standard CNN-based baselines: MiB and PLOP and compared them against NeST.

Table 8: Additional performance comparison on Pascal VOC under different CNN-based architecture baselines for *overlapped* setting. Best results are marked in **Bold**.

| Methods | 15-5 (2 tasks) | | | 15-1 (6 tasks) | | |
|---|---|---|---|---|---|---|
| | 1-15 | 16-20 | All | 1-15 | 16-20 | All |
| **MiB** | | | | | | |
| Baseline Cermelli et al. (2020) | 76.8 | 49.1 | 70.2 | 45.2 | 15.7 | 38.2 |
| MiB + NeST Xie et al. (2024) | 77.1 | 50.1 | 70.7 | 61.7 | **20.4** | 51.8 |
| MiB + Ours | **78.7** | **52.1** | **71.2** | **65.8** | 18.7 | **60.6** |
| **PLOP** | | | | | | |
| Baseline Douillard et al. (2021) | 77.0 | 50.9 | 70.8 | 66.8 | 22.3 | 56.2 |
| PLOP + NeST Xie et al. (2024) | 77.6 | 55.8 | 72.4 | 72.2 | **33.7** | 63.1 |
| PLOP + Ours | **78.1** | **57.2** | **75.3** | **75.1** | 32.4 | **70.3** |

As shown in Table 8, our method consistently outperforms on these CNN baselines. Particularly, on the 15-1 setting, our strategy improves over standard MiB by $\approx$20% on all classes, and improves over PLOP by $\approx$15%.

## D.6 STABILITY OVER LONGER TASK SEQUENCES.

We analyze the stability of our proposed approach over longer task sequences. We adopt the ADE20K dataset with the 100-5 (11 steps) and 100-10 (6 steps) settings, which are rigorous tests for long-term stability. As seen in Fig. 10a (100-5) and Fig. 10b (100-10) settings, performance remains robust across the incremental sequence. In the 100-5 setting, after the initial drop, the model stabilizes, and its performance consistently hovers in the 32%- 37% range.

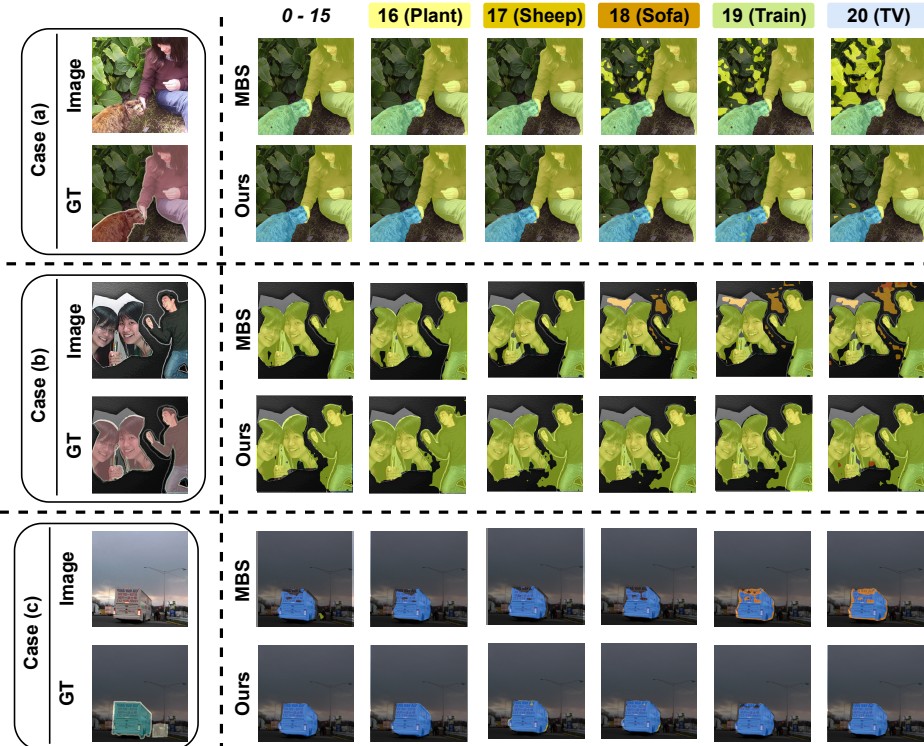

Figure 11: Additional visual comparison on the $15 - 1$ setting of the Pascal VOC between MBS Park et al. (2024) and ours.

## D.7 QUALITATIVE RESULTS.

We provide the qualitative results of our proposed approach when compared with Park et al. (2024) in Fig. 11. We observe that Park et al. (2024) partly maintains knowledge about the object learned previously; however, as the new incremental tasks come along, it starts producing many false positives. As the task progresses, these false positives also get denser. On the other hand, our proposed approach maintains the learning of previous classes while adapting to new knowledge in a more effective manner. For example, in case(b), the sheep is learned in the third task and is maintained thereafter. While Park et al. (2024) partly maintains knowledge learned from the fundamental task and carries it forward.

# E ADDITIONAL ABLATION STUDIES AND SENSITIVITY ANALYSIS

In this section, we perform an additional ablation study to analyze the effectiveness of previously learned similar classes, key hyperparameters, and scaling for a real-world scenario. We perform this study on the Pascal VOC dataset in the overlapped scenario for different settings.

### E.1 Effectiveness of class similarity $\mathcal{C}_s$.

In this study, we evaluate the effectiveness of using previously learned similar classes. To validate our hypothesis, we transfer knowledge from previous dissimilar classes ($\mathcal{C}_{ds} = \mathcal{C}_{0:t-1} - \mathcal{C}_s$) instead of the similar. We perform this ablation on the Pascal dataset with $15 - 1$ and $15 - 5$ settings.

Table 9: Ablation study on Pascal VOC to study the impact of similar $\mathcal{C}_s$ and dissimilar $\mathcal{C}_{ds}$ classes on the complete pipeline. **Bold** denotes the best result.

| $\mathcal{C}_s$ | $\mathcal{C}_{ds}$ | 15-1 (6 tasks) | | | 15-5 (2 tasks) | | |
|---|---|---|---|---|---|---|---|
| | | 1-15 | 16-20 | All | 1-15 | 16-20 | All |
| ✗ | ✓ | 78.5 | 36.6 | 71.5 | 79.6 | 66.8 | 76.4 |
| ✓ | ✗ | **83.3** | **72.0** | **80.5** | **83.9** | **76.0** | **81.6** |

As seen in Table 9, $\mathcal{C}_{ds}$ significantly impacts the performance of both fundamental and incremental classes by a significant margin. Since dissimilar classes contain entirely different contextual information, their representations fail to provide meaningful information. These results confirm that similar class representations serve as a more effective foundation, thus enabling the model to adapt to new classes while preserving the knowledge of previously learned ones.

Table 10: Ablation study (Pascal VOC) on different similarity metrics. **Bold** denotes the best result.

| Metrics | 15-1 (6 tasks) | | | 15-5 (2 tasks) | | |
|---|---|---|---|---|---|---|
| | 1-15 | 16-20 | All | 1-15 | 16-20 | All |
| Cosine | 80.8 | 60.4 | 76.5 | 83.4 | 72.8 | 81.1 |
| Manhattan | 78.5 | 44.9 | 71.2 | 83.2 | 73.7 | 81.3 |
| MSE | 70.9 | 66.4 | 75.6 | 83.6 | 72.3 | 80.7 |
| **Euclidean** | **83.3** | **72.0** | **80.5** | **83.9** | **76.0** | **81.6** |

Table 11: Ablation study (Pascal VOC) on identifying the most appropriate $\sigma$. **Bold** denotes the best result.

| $\sigma$ | 15-1 (6 tasks) | | | 15-5 (2 tasks) | | |
|---|---|---|---|---|---|---|
| | 1-15 | 16-20 | All | 1-15 | 16-20 | All |
| 0.01 | 79.7 | 63.7 | 75.3 | 83.4 | 73.7 | 79.8 |
| 0.03 | 78.0 | 40.1 | 69.7 | 83.2 | 74.3 | 80.1 |
| **0.05** | **83.3** | **72.0** | **80.5** | **83.9** | **76.0** | **81.6** |
| 0.07 | 80.7 | 56.7 | 73.9 | 83.3 | 73.5 | 79.6 |
| 0.09 | 78.8 | 46.6 | 68.9 | 83.7 | 73.5 | 79.9 |

### E.2 Effectiveness of similarity metrics.

We perform this study to analyze the effectiveness of using Euclidean distance as an efficient metric to identify the similarity between learned class tokens and deviated class embeddings. Table 10 presents our findings.

A small distance implies a high semantic similarity. While other metrics like cosine similarity focus on orientation, Euclidean distance considers both magnitude and orientation, providing an appropriate measure of geometric distance in the latent space. As observed, the Euclidean distance consistently outperforms the other metrics, particularly for incremental tasks.

### E.3 Determining the appropriate $\sigma$.

We perform this study to determine the most appropriate $\sigma$ in the gaussian noise $\mathcal{N}$ in §3.4. Table 11 presents our findings. Standard deviation ($\sigma$) of the Gaussian noise plays a crucial role during knowledge transfer. We conduct an ablation study and found that $\sigma$=0.05 consistently yields the best performance across both tasks. This value strikes as an effective trade-off. Smaller values (*eg* 0.01) fail to diversify the initialization meaningfully, while larger values (*eg* 0.10) inject excessive noise.

### E.4 Analyzing the issue of scalability.

Adapting to new classes may require a large number of images, which might not be possible in a real-world scenario. To this end, we perform an ablation, visualized in Fig. 12, to analyze whether the influence of the selective context can reduce the number of images in the new incremental tasks. To this end, we employ 50% and 25% of the total number of images in the incremental setting to

Table 12: Ablation study on Pascal VOC to analyze the effectiveness of $\alpha$. **Bold** denotes the best result.

| $\alpha$ | 15-1 (6 tasks) | | | 15-5 (2 tasks) | | |
|---|---|---|---|---|---|---|
| | 1-15 | 16-20 | All | 1-15 | 16-20 | All |
| 0.1 | 82.1 | 67.7 | 79.1 | 82.2 | 73.4 | 80.7 |
| 0.3 | 82.4 | 49.2 | 75.0 | 83.3 | 73.6 | 81.5 |
| 0.5 | 78.8 | 57.8 | 74.4 | 82.3 | 72.1 | 80.3 |
| 0.7 | 81.9 | 67.4 | 78.9 | 82.4 | 72.6 | 80.6 |
| **0.9 (Ours)** | **83.3** | **72.0** | **80.5** | **83.9** | **76.0** | **81.6** |

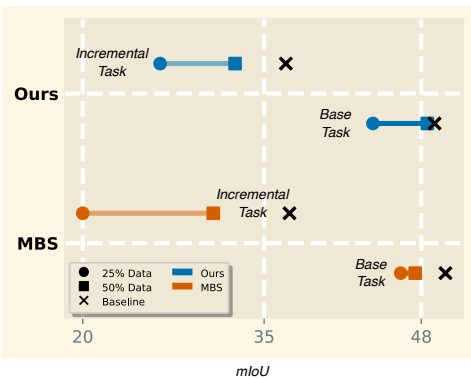

Figure 12: Ablation study on analyzing the issue of scalability on ADE20K "$100 - 50$" setting.

validate our hypothesis. To align closely with real-world scenarios, we chose the ADE20K dataset on "$100 - 50$", an overlapped setting. We also compare our approach to the recent method Park et al. (2024) with similar settings for fair comparison.

Upon training the complete task, with 50% incremental data, we find that our approach (■) aligns closely with the baseline (where complete data was used "**x**"), while for Park et al. (2024), there appears some distance, This depicts that leveraging less data may hinder performance but the trade-off between performance and quantity is manageable. We observe a similar trend for 25% incremental data as well. The difference in performance as compared to baseline is *less* in ours compared to Park et al. (2024). This analysis shows that, with minimal performance drop, this approach can be implemented in a real-world scenario by accommodating a minimal new dataset for incremental learning.

## E.5 EFFECTIVENESS OF THE $\alpha$.

In § 3.4, we discuss about the Context Transfer Attention (CTA) and we add a controlled noise component $\mathcal{N}$ (see Eqn. 5) to provide a distinction between previously learned similar classes $\mathcal{C}_s$ and the new class $\mathcal{C}_t$. To maintain the balance between $\mathcal{C}_s$ and $\mathcal{N}$, we include a weighting hyperparameter $\alpha$. In Table 12, we analyze this hyperparameter by providing different values.

From the table, it is evident that our proposed strategy outperforms the alternative values for the $\alpha$. Particularly, when $\alpha$ is low (*eg* 0.1), the representation is more influenced by the noise component, introducing flexibility. Conversely, when $\alpha$ (*eg* 0.9) is high, the representation remains close to the original, offering more guidance. In the intermediate range, the representation shows the ambiguous behaviour that is neither sufficiently guided nor flexible, which explains the observed performance dip. This observed trend, however, shows the inconsistency, yet reflects the underlying conditions of the $\alpha$ trade-off.

## E.6 FINE-TUNING FREQUENCY THRESHOLD $\varepsilon$ AND MARGIN $M$

In this study, we analyze the effect of other hyperparameters like Margin $M$ (Table 13) and the frequency threshold $\varepsilon$ (Table 14). We perform this analysis for Pascal-VOC on both "15-1" and "15-5" settings.

The $\mathcal{L}_{ct}$ margin dictates how strictly the model forces separation between the new class and similar old classes. At $M = 0.0$, we observe the lowest performance, as it fails to enforce a distinct decision boundary. While at $M = 1.0$, we observe the highest performance. A larger margin provides a stronger constraint, compelling the model to push the new class representation sufficiently far away from prior similar classes.

For the frequency threshold, our experiments consistently show that 0.15 is the optimal value. When we increased the threshold to 0.30 or 0.45, the performance declined. Including a larger pool of similar classes (top 45%) introduces semantic dilution. A tighter threshold (0.15) ensures that we only transfer knowledge from highly relevant predecessors, preventing confusion for the new class.

Table 13: Fine-tuning Margin $M$ in Eq. (6). **Bold** denotes the best result.

| Margin $M$ | 15-1 (6 tasks) | | | 15-5 (2 tasks) | | |
|---|---|---|---|---|---|---|
| | 1-15 | 16-20 | All | 1-15 | 16-20 | All |
| 0.0 | 75.6 | 34.8 | 70.0 | 82.4 | 72.2 | 78.7 |
| 0.5 | 75.8 | 45.2 | 74.6 | 82.7 | 75.4 | 79.8 |
| **1.0** | **83.3** | **72.0** | **80.5** | **83.9** | **76.0** | **81.6** |

Table 14: Fine-tuning threshold $\varepsilon$ in § 3.3.2. **Bold** denotes the best result.

| Threshold $\varepsilon$ | 15-1 (6 tasks) | | | 15-5 (2 tasks) | | |
|---|---|---|---|---|---|---|
| | 1-15 | 16-20 | All | 1-15 | 16-20 | All |
| 0.45 | 60.9 | 33.1 | 55.4 | 83.2 | 72.4 | 77.2 |
| 0.30 | 61.2 | 48.1 | 59.5 | 83.3 | 74.0 | 79.7 |
| **0.15** | **83.3** | **72.0** | **80.5** | **83.9** | **76.0** | **81.6** |

### E.7 SENSITIVITY ANALYSIS OF HYPERPARAMETERS

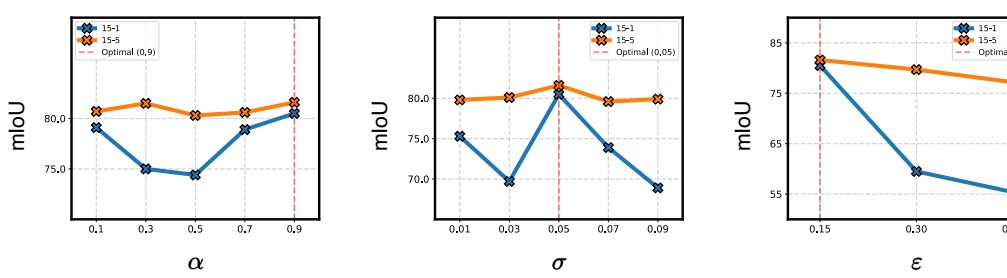

Figure 13: Analyzing the sensitivity of different hyperparameters across Pascal-VOC, 15-1 and 15-5 incremental settings.

To evaluate the robustness of our proposed method, we conduct a comprehensive sensitivity analysis on the key hyperparameters: the perturbation noise standard deviation $\sigma$, the noise trade-off $\alpha$, and the frequency threshold $\varepsilon$. We analyze the impact of these parameters across distinct experimental settings—the long-sequence 15-1 (6 tasks) and the short-sequence 15-5 (2 tasks).

The noise standard deviation $\sigma$ in Eq. (5) controls the magnitude of the gap between the new class initialization and its source priors. As shown in Fig. 13 and Table 10, the optimal value is 0.05 for both the extensive 15-1 setting (6 tasks) and the shorter 15-5 setting (2 tasks). In 15-5 (Short Sequence), the method is highly robust. Across the entire range of $\sigma$ (0.01 to 0.09). While in 15-1 (Long Sequence), we observe a distinct spot at 0.05.

For the noise trade-off $\alpha$ (in Table 11), the optimal value is 0.9 for both settings. Lower $\alpha$ values (0.1 - 0.5) introduce too much randomness. The stability of the method is anchored in the prior knowledge; therefore, sensitivity to low $\alpha$ is a confirmation that the prior knowledge is necessary.

In the frequency threshold ($\varepsilon$), our ablation (in Table 14) shows 0.15 is optimal. Increasing this includes noisy, dissimilar classes, which naturally degrade performance. This is not sensitivity to a parameter, but sensitivity to data quality.

## F LIMITATIONS AND FUTURE WORK

In this work, we focus on utilizing previously learned knowledge in an adaptive manner and selectively transferring knowledge from classes that resemble the new class. Although our method identifies the subset of most similar classes that resemble the new class, further improvement is required to maintain the trade-off between the forward and backwards knowledge transfer (impact on new and old classes, respectively). We believe that identifying more optimal approaches to leverage previous knowledge can provide better direction for the CISS problem in real-world scenarios.

# G USAGE OF LLMS

We have used the LLMs strictly to aid in polishing and improving grammatical mistakes. There is no other usage of LLMs involved in our work.

