# OpenReview forum: "SELECT: SELEctive Context Transfer for Class Incremental Semantic Segmentation"
_ICLR.cc/2026/Conference — Submitted to ICLR 2026_

### Official Review · Reviewer_ReFa · 2025-10-27

**Soundness:** 3
**Presentation:** 3
**Contribution:** 3
**Rating:** 4
**Confidence:** 4

**Summary:**

This paper proposes SELECT, a method for Class-Incremental Semantic Segmentation (CISS) that addresses catastrophic forgetting through selective knowledge transfer. The key idea is to identify semantically similar classes from previous tasks and use Context Transfer Attention (CTA) to selectively transfer their knowledge when learning new classes. The method measures representational perturbation to identify similar classes, then uses an attention mechanism to aggregate knowledge from these classes. A context transfer loss enforces separation between new and source class representations. Experiments on PASCAL VOC and ADE20K show improvements over recent methods.

**Strengths:**

## 1. Well-motivated
* The selective knowledge transfer approach based on inter-class similarity is intuitive and differentiates from prior work that either uses background initialization (MiB, SSUL) or transfers from all classes indiscriminately (NeST)
* The similarity metric (Eq. 1) offers a novel way to measure semantic alignment between old and new classes
* The combination of CTA with controlled noise (Eq. 5) and discriminative loss is a reasonable approach to balance knowledge transfer and separation

## 2. Well-written
* Well-structured paper with clear motivation and problem formulation
* Figure 3 effectively illustrates the overall framework
* The writing is generally clear

**Weaknesses:**

## 1. Unfair Comparison with NeST
The paper compares SELECT (using ViT-B/16 backbone, stated in Section 4.1) against NeST baselines that appear to use the Swin-B backbone. According to the original NeST paper, ViT-B achieves 76.5 mIoU (All classes) on VOC 15-1, significantly higher than the 71.4-72.2 reported in Table 2. The reported NeST scores in Table 2 match the Swin-B results from the original paper. This creates an unfair comparison where SELECT (ViT-B) is compared against a weaker baseline (NeST with Swin-B), potentially overstating the contribution.

## 2. Substantial Reproduction Gaps for Main Baseline, MBS
The reproduced MBS results differ significantly from reported values, especially on the ADE20K dataset:
* VOC 15-1: 80.6 (reported) → 79.0 (reproduced)
* ADE20K 100-5: 42.8 (reported) → 38.8 (reproduced)

While reproduction discrepancies can occur due to various factors (random seeds, library versions, hardware differences), the paper would benefit from providing more detailed information about the reproduction process:

* Environmental setup details (PyTorch/CUDA versions, GPU specifications)
* Number of random seeds used and variance observed
* Whether reproduction was verified with the original authors
* Discussion of potential sources of discrepancy

This additional information would strengthen confidence in the experimental comparisons and help the community better understand and reproduce the results.

## 3. Unvalidated Core Assumption
* Lines 279-280 state: "A small deviation between these two vectors indicates that the content of the image xi is contextually aligned with class k."
* This is the foundation of the method, yet this critical assumption lacks proper validation: The similarity metric (Eq. 1) is foundational to the entire method, yet Figure 4 only shows that perturbations vary across classes without demonstrating correlation with actual semantic similarity.
* The paper needs to demonstrate that this metric reliably captures semantic similarity (e.g., with empirical or statistical analysis)

## 4. Sensitive Hyperparameters
The method introduces several hyperparameters that appear to have a substantial impact on performance: $\sigma$ (Table 9), $\alpha$ (Table 10), and $L_{ct}$.
These sensitive hyperparameters may suggest the method is brittle and may be difficult to tune for new datasets or scenarios.

Please provide the sensitivity analysis for the $L_{ct}$ and threshold value.

**Questions:**

## 1. Aggressive Language
The Phrases below are unnecessarily strong for academic writing. They could be softened while still conveying the limitations being addressed.

- Line 52: "This approach is fundamentally misguided ..."
- Line 80: "this brute-force-like strategy ..."
- Line 81: "This is akin to learning a new skill by reading an entire library instead of consulting a few expert texts"

---

> ### Author Response · Authors · 2025-11-24
> **Weaknesses 1: Unfair Comparison with NeST (Part 1 of 5)**
>
> We originally reported the Swin-B results for NeST to align with the primary results highlighted in the original NeST paper. However, we acknowledge this created ambiguity in Table 2 regarding the specific backbone usage.
>
> To address this and demonstrate the robustness of our method, we have conducted a direct comparison for ViT-B Backbone. As shown in the table below, our method (SELECT) outperformed NeST on almost all task sequences.
> | Models | Backbone | 15-1 | 15-5 | 10-1 |
> | -------- | -------- | -------- | -------- | -------- |
> | Incrementer     | ViT-B     | 75.5     | 79.9 | 70.2|
> | MiB     | ViT-B     | 53.5     |80.2|25.5|
> | NeST+MiB     | ViT-B     | 76.5     |80.3|**71.9**|
> | **Ours**     | **ViT-B**    | **80.5**|**81.6**|70.4|
>
> - Consistent Superiority: Even against the ViT-B baseline (76.5%), our method achieves 80.5%, maintaining a substantial +4.0% margin on the critical 15-1 task.
> - Broad Improvement: Our method consistently outperforms other ViT-based approaches, including Incrementer and the standard MiB.
>
> *We have updated Table 2 to explicitly add a column specifying the Backbone for every method. We have also added the above table in the appendix (section D.3) to provide a transparent comparison for future benchmarking.*

---

> ### Author Response · Authors · 2025-11-24
> **Weaknesses 2: Substantial Reproduction Gaps for Main Baseline, MBS (Part 2 of 5)**
>
> We confirm that we observed this discrepancy between the results reported in the original MBS paper and those we obtained from our reproduction, particularly on the complex ADE20K dataset. Below, we provide the details of our reproduction process as requested.
>
> ### Reproduction Process and Environmental Setup:
> Our primary goal was to establish a fair and consistent experimental framework for all methods.
> - Our reproduced MBS results were obtained by running the official, publicly available source code provided by the MBS authors. We used their implementation without modifying the core architecture or loss functions.
> - To ensure a true comparison, our proposed method (SELECT) and MBS were trained and evaluated within the exact same, unified environment.
> - The environmental details are as follows:
>    - PyTorch Version: 1.10.1
>    - CUDA Version: 11.1
>    - GPU: NVIDIA A100, single GPU (unlike the original work, which uses multiple GPUs)
>    - Backbone: ViT-B pre-trained on ImageNet
>
> ### Multiple runs and Variance
> - As also mentioned in the manuscript ($\S 4.2$), the reproduced scores are the average mIoU values obtained from multiple runs. The average variance observed for MBS was $\pm 0.57$ on VOC and $\pm 0.43$ on ADE20K. We have updated our results tables to include separate variations for all tasks, making this information clearer (removing the percentage increase or decrease).
>
> ### Potential Sources of Discrepancy
> - We consulted with the authors of MBS, requesting clarification on the discrepancy in results, and they responded that it could be due to batch size or other environmental conditions. We agree that we did not verify the reproduced results with the authors, which we will do after the review period.
>
> *We have also added an additional section in the Appendix (section D.1) to include this analysis.*

---

> ### Author Response · Authors · 2025-11-24
> **Weaknesses 3: Unvalidated Core Assumption (Part 3 of 5)**
>
> ### Theoretical Mechanism: Why deviation measures semantic similarity
> - Our method leverages the properties of a well-trained representational space where features are not random but semantically clustered.
>    - The mechanism: When we compute the interaction between an incremental class image $x_i$ ("Sheep") and a base class token $e_{k}$ ("Dog").
>       - Scenario A (semantically similar): The sheep image contains visual features (snout, four legs) that are compatible with the Dog token. The attention mechanism finds strong alignment, resulting in a refined token $\hat{e}$ that remains close to the original $e_{k}$ in the representational space. The deviation (Euclidean distance) is therefore small.
>      - Scenario B (Semantically dissimilar): If we pair the same “Sheep” image with a dissimilar token $e_{k}$ (e.g. "Car"), the image features actively conflict with the token’s expected features (metal, wheels). This context mismatch forces the resulting representation $\hat{e}$ to drift significantly from $e_{k}$. This deviation is therefore large.
> ### Empirical evidence: Correlation heatmap
> - To validate this claim, we generated a class similarity correlation heatmap (presented in the Appendix).
>    - Setup: The X-axis represents Base classes and the Y-axis represents new incremental classes. The cell color represents the calculated similarity metric (inverse of deviation)
>    - Results: The heatmap reveals distinct, semantically coherent clusters rather than random noise. Particularly, the incremental class "Sheep" shows a high similarity score with base classes like "Cow", "Horse", "Dog", and "Cat", but a near-zero score with "Aero" or "Motor".
>
> *We have provided a correlation heatmap in the appendix (section C.2), comparing the new incremental class with all the previous classes.*

---

> ### Author Response · Authors · 2025-11-24
> **Weaknesses 4: Sensitive Hyperparameters (Part 4 of 5)**
>
> While the method involves multiple hyperparameters, our analysis of Tables 9 and 10 (Now Table 11 and 12 respectively) reveals two critical insights:
>
> ### Parameter Consistency (Disproving Brittleness)
> A method is "brittle" if the optimal hyperparameters change drastically when the task changes (e.g., requiring $\sigma=0.01$ for one task but $\sigma=0.09$ for another).
> Our results show the exact opposite:
> - For $\sigma$ (Table 11): The optimal value is 0.05 for both the extensive 15-1 setting (6 tasks) and the shorter 15-5 setting (2 tasks).
> - For $\alpha$ (Table 12): The optimal value is 0.9 for both settings.
> - The method is generalizable. The same hyperparameters work effectively regardless of the increment steps, suggesting they capture fundamental properties of the feature space rather than overfitting to a specific scenario.
> ### Sensitivity vs. Task Difficulty
> The analysis shows that sensitivity is directly correlated with the length of the continual learning sequence, which is expected behavior for any CL method.
> - Sensitivity to Standard Deviation ($\sigma$)
>    - In 15-5 (Short Sequence): The method is highly robust. Across the entire range of $\sigma$ (0.01 to 0.09), the 'All' mIoU only fluctuates by $\approx$ 2%.
>    - In 15-1 (Long Sequence): We observe a distinct spot at 0.05.
>       - Too low (< 0.03): The "gap" is too small, leading to symmetry with old classes and forgetting.
>       - Too high (> 0.07): The noise destroys the semantic guidance, leading to random initialization.
>       - Optimal (0.05): This value represents the ideal separation margin in the embedding space. The fact that it peaks sharply in the difficult 15-1 task confirms it is doing exactly what it is designed to do: navigating a tight optimization landscape.
> - Sensitivity to Alpha ($\alpha$)
>    - The performance favors higher $\alpha$ (0.9), meaning we retain 90% of the original semantic signal.
>    - Analysis: Lower $\alpha$ values (0.1 - 0.5) introduce too much randomness, effectively discarding the "warm start" benefit of our method. The stability of the method is anchored in the prior knowledge; therefore, sensitivity to low $\alpha$ is a confirmation that the prior knowledge is necessary, not a sign of weakness.
> - Sensitivity to $L_{ct}$ and Threshold ($\varepsilon$)
>    - $L_{ct}$ (Loss Weight): As with $\sigma$, the margin hyperparameter shows a consistent preference for 1.0 across tasks, providing a stable baseline for separation.
>    - Threshold ($\varepsilon$): The study shows 0.15 is optimal. Increasing this includes noisy, dissimilar classes, which naturally degrades performance. This is not sensitivity to a parameter, but sensitivity to data quality.

---

> ### Author Response · Authors · 2025-11-24
> **Question 1: Aggressive Language (Part 5 of 5)**
>
> We agree that the language in these sections was stronger than necessary. We have revised the manuscript (and highlighted the lines) with more formal terminology while retaining the technical meaning. Particularly:
> - Line 52: "This approach is fundamentally misguided ...": "This approach may be suboptimal ..."
> - Line 80: "this brute-force-like strategy ...": "this strategy ..."
> - Line 81: "This is akin to learning a new skill by reading an entire library instead of consulting a few expert texts": **Removed**
>
>
> We thank the reviewer for this suggestion

---

### Official Review · Reviewer_7RG7 · 2025-10-28

**Soundness:** 2
**Presentation:** 2
**Contribution:** 2
**Rating:** 2
**Confidence:** 4

**Summary:**

This paper proposes a method for class-incremental semantic segmentation (CISS) by transferring only semantically relevant prior knowledge from a small set of past classes to each new class rather than relying on background or indiscriminative transfer. The authors propose to identify similar past classes via a representational perturbation test and aggregate their tokens with a context transfer attention (CTA) module. They also introduce a consensus-based class selection, which, for each new-task image, picks the past class with minimum token perturbation. It then keeps only past classes that frequently win across the dataset. The CTA module forms a guided token by attending over tokens of the selected prior classes, where query is the mean context with softmax weighting. This gives initialization for a new class. Controlled perturbation is added to the CTA token to create a buffer zone in representation space and mitigate overlap with its sources, and a margin-based context transfer loss to explicitly separate new and source classes. Experiments on standard datasets show consistent improvements across disjoint and overlapped scenarios and incremental learning settings over strong baselines. Detailed ablations are given for the data selection, CTA, noise, and loss terms.

**Strengths:**

This paper provides a nice motivational discussion on why background-centric or global transfer is suboptimal. It then proposes to use per-class selective transfer guided by a concrete, measurable stability criterion. The token selection, CTA, noise and margin loss are conceptually simple and easy to integrate. Perturbation and context transfer loss explicitly enforce separation from source classes which addresses a common source for forgetting when transferring from semantically close classes. Experiments are done on various overlapped settings, and detailed ablations make the contribution of each part convincing.

**Weaknesses:**

Data selection is based on token perturbation, but its robustness under shift or noise is not clearly explained. Experiments focus on ViT with a Segmenter-like head; it’s unclear whether selection and CTA behave similarly for other architectures. There are many hyperparameters like frequency threshold, noise mix and margin need to be fine-tuned. It is not clear to me how the proposed method addresses the specific problem of background shift in CISS.

**Questions:**

1 What is the per-step cost of computing perturbations for all past classes, CTA calculation and context transfer loss? How does complexity of the proposed method scale to larger models?
2. Do the performance gains hold for other model architecture and longer continual learning task sequences? Does drifting and instability happen as the number of *selected* prior classes grows over time?

---

> ### Author Response · Authors · 2025-11-24
> **Weaknesses 1: Robustness in token perturbation (Part 1 of 5)**
>
> We clarify that the token perturbation is not merely a selection heuristic, but rather a robustness filter designed to explicitly handle such noise.
> ### Robustness through Aggregation:
> - Our selection works on class-level tokens. These tokens represent high-level semantic prototypes aggregated over the entire dataset.
>    - Low-level input noise or temporary domain shifts are largely averaged out during the feature extraction and pooling process. The tokens effectively act as denoised semantic anchors, making the selection process inherently stable against input perturbations.
> ### Robustness under shift/noise:
> - The reviewer asks about robustness under shift/noise. Crucially, our perturbation step simulates this very noise.
> - By artificially adding noise to the tokens, we observe the prediction consistency.
> - If a class relationship is fragile (i.e. sensitive to noise/shift), the perturbation will cause the prediction to flip, and our algorithm will automatically reject that class as a candidate for transfer.
> - Therefore, the method does not suffer from noise; it uses simulated noise to filter out unstable connections, ensuring we only transfer knowledge from classes with robust, structurally sound relationships.
> ### Consensus-based Filtering:
> - As mentioned in our implementation details we use a consensus method using frequency threshold. We do not select prior based on a single noisy sample. A class is selected only if it appears as a semantic neighbour across a statistically significant portion of data distribution (15%). This eliminates outliers caused by transient distribution shifts.

---

> ### Author Response · Authors · 2025-11-24
> **Weaknesses 2 and Question 2: Performance gains on other architectures and longer task sequences (Part 2 of 5)**
>
> Below we provide analysis that generalizes effectively to CNN architectures and maintains stability over long task sequences.
> ### Generalization to Other CNN Architectures
> - To verify that our gains are not specific to a single architecture (Transformers), we integrated our method into two standard CNN-based CL baselines: MiB and PLOP. We also compared this against NeST.
> - Improvement over Baseline: On the 15-1 setting, our strategy improves over standard MiB by $\approx$20\% on all classes, and improves over PLOP by $\approx$15\%.
> - Improvement over NeST: Our method outperforms NeST in total mIoU across both settings, demonstrating that our proposed strategy are architecture-agnostic modules and can boost across various backbones.
>
> | Model | 15-1 |  |15-5 |  |
> | -------- | -------- | -------- |-------- | -------- |
> |      | 1-15     | 16-20     |1-15     | 16-20     |
> | MiB     | 45.2     | 15.7     |76.8     | 49.1     |
> | MiB+NeST | 61.7     | **20.4**     |77.1     | 50.1     |
> | MiB+Ours | **65.8**     | 18.7     |**78.7**     | **52.1**     |
>
> | Model | 15-1 |  |15-5 |  |
> | -------- | -------- | -------- |-------- | -------- |
> |      | 1-15     | 16-20     |1-15     | 16-20     |
> | PLOP     | 66.8     | 22.3     |77.0     | 50.9     |
> | PLOP+NeST | 72.2     | **33.7**     |77.6     | 55.8     |
> | PLOP+Ours | **75.1**     | 32.4     |**78.1**     | **57.2**     |
>
> ### Stability in Longer Task Sequences
> - We address the validity of stability in longer task sequences using the ADE20K dataset with the 100-5 (11 incremental steps) and 100-10 (6 incremental steps) setting, which are rigorous tests for long-term stability.
> - Our analysis shows that significant drift does not occur. As seen in Tables below, performance remains robust across the sequence.
>    - In the 100-5 setting (11 steps), after the initial drop (typical when moving from base to incremental), the model stabilizes.
>    - It does not exhibit an instability characteristic of concept drift.
> - The stability is directly attributed to our Consensus-based Class Selection.
>    - Filtering Outliers: Our selection strategy ensures that we only use prior classes that are semantically relevant.
>    - By enforcing this, we prevent noisy or irrelevant classes from being used for initialization. This ensures that the transfer remains effective and does not introduce the drift.
>
> | Model |  |  | | | |100-5 ||||||
> | -------- | -------- | -------- |-------- | -------- |-------- |-------- |-------- |-------- |-------- |-------- |-------- |
> |      | 1-100     | 101-105     |101-110     | 101-115     | 101-120 | 101-125 |101-130|101-135|101-140|101-145|101-150|
> | Ours     | 46.7    |22.6|31.6|32.8  |36.5| 36.3 |37.3|36.4|33.1|28.7|27.5|
>
> | Model |  |  | 100-10| | | |
> | -------- | -------- | -------- |-------- | -------- |-------- |-------- |
> |      | 1-100     | 101-110     |101-120     | 101-130     | 101-140 | 101-150 |
> | Ours     | 48.0    |30.2   |41.1  | 41.8   | 39.5 | 34.8 |
>
> *We have added these detailed task-wise breakdowns and the CNN experiments to the appendix (sections D.5 and D.6 respectively) as well.*

---

> ### Author Response · Authors · 2025-11-24
> **Weaknesses 3: Fine-tuning of hyperparameters (Part 3 of 5)**
>
> We acknowledge that our method involves multiple hyperparameters. However, we emphasize that these values were not chosen arbitrarily but were determined through rigorous ablation studies to ensure optimal performance. Furthermore, our analysis shows that the behaviour of these hyperparameters is consistent and interpretable.
> We provide below the detailed analysis for the same:
> ### Frequency threshold:
> - Settings explored: We compared three threshold values of 0.15, 0.30 and 0.45.
> - Optimal value: 0.15 (top 15% most similar classes)
> - Analysis: Our experiments consistently showed that 0.15 is the optimal value.
>    - When we increased the threshold to 0.30 or 0.45, performance declined.
>    - Conclusion: Including a larger pool of similar classes (top 45%) introduces semantic dilution. Less relevant classes get mixed into the initialization, introducing noise. A tighter threshold (0.15) ensures that we only transfer knowledge from highly relevant predecessors, preventing confusion for the new class.
>
> | Frequency threshold | 15-1 |  |15-5 |  |
> | -------- | -------- | -------- |-------- | -------- |
> |      | 1-15     | 16-20     |1-15     | 16-20     |
> | 0.45     | 60.9  | 33.1    |83.2    |72.4     |
> | 0.30 |61.2 | 48.1  |83.3   |74.0  |
> |**0.15** | **83.3**  | **72.0**  |**83.9**     |**76.0** |
>
> ### Noise Mix:
> - Analysis: As detailed in Table 10 (now Table 12) in the appendix, we analyzed the effect of adding a controlled noise component to the initialization.
> - Reasoning: The parameter effectively controlled the learning of new classes while retaining the previously learned similar classes.
>    - High Noise: The representation becomes too random/flexible, losing the benefit of the warm start.
>    - Low Noise:  The representation offers more guidance, while also leaving some room for learning new classes
> ### Context Transfer Loss Margin:
> - Settings Explored: We conducted an ablation study with margin values of 0.0, 0.5 and 1.0.
> Optimal Value: 1.0
> - Analysis: The margin dictates how strictly the model forces separation between the new class and similar old classes.
>    - M=0.0: Provided the lowest performance, as it fails to enforce a distinct decision boundary.
>    - M=1.0: Yielded the highest performance.
>    - Conclusion: A larger margin enforces a stronger constraint, compelling the model to push the new class representation sufficiently far away from prior similar classes. This results in clearer decision boundaries, validating our choice of 1.0.
>
> | Margin | 15-1 |  |15-5 |  |
> | -------- | -------- | -------- |-------- | -------- |
> |         | 1-15  | 16-20     |1-15     | 16-20     |
> | 0.0     |  75.6  | 34.8    |82.4    |72.2    |
> | 0.5     | 75.8   | 45.2   |82.7 | 75.4 |
> | **1.0** | **83.3**   | **72.0**    |**83.9**     |**76.0** |
>
> *We have included these ablation details with relevant explanation in the appendix (section E.6) as well.*

---

> ### Author Response · Authors · 2025-11-24
> **Weaknesses 4: Addressing Background shift in CISS (Part 4 of 5)**
>
> Background shift occurs when pixel classification of a future class (labelled as background in previous steps) are ambiguous. Our method addresses this using the noise perturbation and Context Transfer Loss. By explicitly enforcing a controlled noise and a separation margin in the loss, we create a "buffer zone" in the representation space. This prevents the new class from being misclassified and collapsing into the background/old-class representations.
>
> While methods like MBS address this by initializing the new class from the background classifier, we address the shift effectively through a different, foreground-centric approach:
> ### Overcoming background suppression via foreground-centric approach:
> - The core of the background shift problem is that a randomly initialized new classifier produces low activations for the new class, while the old model produces high activations for the background. This makes it difficult for the new class to stand out from the background.
> - Our method solves this by initializing the new class with the weights of a semantically similar old class
>    - Since the model has already learned to distinguish between dogs and cats (foreground) and the background, transferring these weights gives sheep classifiers better initialization against the background.
>    - The new class does not start as undefined background. It starts with a strong foreground prior.
> ### Leveraging Learned Foreground-Background Separability
> - By transferring knowledge from a similar old class, we also transfer the learned boundary between that old class and its background.
> - Because sheep and dogs (and cats) share similar visual features, the features that separate dogs from the background are largely the same features that separate sheep from the background.

---

> ### Author Response · Authors · 2025-11-26
> **Question 1: Per-step cost of computation (Part 5 of 5)**
>
> The additional cost of our method is negligible compared to the standard training loop of the backbone model.
> We analyze the cost for each component below.
> ### Cost of CTA & Perturbation (One-Time Initialization Cost)
> - The CTA and Perturbation are not computed per training step. They are computed only once at the beginning of each new incremental task to initialize the weights.
> - This is a simple matrix-vector multiplication with complexity $O(K \cdot D)$. For a standard setup (e.g., $D$ is the hidden dimension, and $K$ are the number of classes), this computation takes milliseconds.
> - Zero impact on per-step training speed.
> ### Cost of Context Transfer Loss (Per-Step Training Cost)
> - The Context Transfer Loss ($L_{ct}$) is computed at each training iteration, but it is extremely lightweight.
> - It calculates the similarity between the current learnable class tokens and the previously learned fixed similar tokens.
> - In our experiments, the training time per epoch with SELECT is identical to the baseline within a margin of error (<1%).

---

### Official Review · Reviewer_EQWq · 2025-10-31

**Soundness:** 3
**Presentation:** 4
**Contribution:** 2
**Rating:** 2
**Confidence:** 4

**Summary:**

The paper introduces a new strategy to select relevant contextual information from the past classes to improve the learning of new classes. It introduces Context Transfer Attention and context transfer loss to select useful information.

**Strengths:**

1. The paper revisits the problems of background leakage and how to leverage past class information when learning new classes. This problem is relevant in incremental segmentation.
2. The paper presentation and motivation are clear.

**Weaknesses:**

1. The Introduction claims that "current methods often fail to leverage prior knowledge effectively". This is not true. Several works such as REMINDER [1*], MBS [Park et al.] already leverage prior knowledge about the background (MBS), or about the relational structures between classes for optimal learning/forgetting trade-off (REMINDER).
2. It's unclear how the perturbation prevents damaging the past knowledge. How does the perturbation "creates a small gap for adapting a new class". And how "creating a small gap for adapting a new class" prevents knowledge loss?
3. The contribution (3) of context transfer loss aims to push away old and new class tokens/weights. This separating loss objective is already introduced in MBS as a form of orthogonality loss. As such, it lacks novelty. The only difference is the similarity function. What are the problems of MBS's orthogonality loss? How the proposed loss fixes those problems? What experiments we need to run to validate those problems and how the proposed loss can fix them?
4. According to Table 2, the performance of proposed method is lower than baselines on two tasks in terms of all metrics. On the other hand, performance gain is minimal (less than 1 p.p) on ADE20k. Overall, the performance gain is minimal. As the accuracy improving is minimal, how is the cost of training SELECT compared with other lightweight techniques such as NeST and INC?
5. What is the core difference when both REMINDER and SELECT aim to learn relevant information from old classes? How is the performance gain compared with similar past-class-learning methods such as REMINDER [1*]?
6. According to Table 4, removing contrastive loss and feature distillation loss yields a higher (all) mIoU on 15-5. It shows that the contribution of contrastive loss is minimal. How shall we interpret this performance drop results?


[1*] Class Similarity Weighted Knowledge Distillation for Continual Semantic Segmentation, CVPR2022.

**Questions:**

Refer to Weaknesses section.

---

> ### Author Response · Authors · 2025-11-24
> **Weaknesses 1 and 5: Distinction with recent works (Part 1 of 5)**
>
> It is true that MBS and MiB leverage background knowledge from past classes to initialize the new class token. However, as mentioned in line 214, the background information consists of an unstructured collection of information of unseen classes. We also observe a similar trend in Fig. 1. The background weights used in MBS provide high-variance initialization that remains unchanged even after training. This information can confuse the model with an unreliable mixed source of initial knowledge rather than targeted guidance.
>
> As for other methods, such as REMINDER and NeST, they leverage knowledge from past classes. The primary focus of REMINDER is to reduce the likelihood of forgetting prior similar classes, regardless of the new class initialisation.
> The gap this paper tries to address is as follows. We notice that a specific type of knowledge that we can find in previously learned classes is not being effectively utilized in the existing methods. Therefore, we try to use semantic relationships with already learned foreground classes to proactively initialize new class classifiers.
> While the cited works are important, they leverage prior knowledge in fundamentally different ways.
>
> - **Distinction from MBS**
>    - **MBS's Method:** It initializes the new class classifier using the weights of the old background classifier.
>    - **Our Method (SELECT):** We do not use the background weights. We argue that leveraging knowledge from semantically similar already learnt foreground classes (e.g., using "dog" and "cat" to help initialize "sheep") provides a much richer and more relevant semantic starting point.
>
> - **Distinction from REMINDER**
> As the reviewer correctly states, REMINDER uses "relational structures" between classes. However, it applies this knowledge reactively.
>    - **REMINDER's Method:** It initializes new class randomly and uses its Class Similarity Weighted Knowledge Distillation loss during training to act as a corrective constraint. This loss forces the model to maintain distinctions between the new, randomly-initialized class and similar old classes.
>   - **Our Method (SELECT):** We use this relational knowledge before training. We initialize the new class classifier by adapting the weights from the most similar past classes.
>    - **Resulting Performance Gain**
> This fundamental difference in approach directly translates to the performance gains observed in our experiments.
>
>
> | Model    | 15-1 |       | 15-5 |       | 19-1     |  |
> | -------- | -------- | -------- | -------- | -------- | -------- | -------- |
> |    | 1-15 | 16-20 | 1-15 | 16-20 | 1-19 | 20 |
> | Reminder     | 68.3         | 27.2   |76.1  | 50.7     |  76.5     | 32.3     |
> |Ours| **83.3**| **72.0**| **83.9**| **76.0**| **81.6** | **68.2**  |
>
> | Model    | 100-10 |       | 100-50 |       | 100-5     |  | 50-50 | |
> | -------- | -------- | -------- | -------- | -------- | -------- | -------- | -------- | -------- |
> |    | 1-100 | 101-150 | 1-100 | 101-150 | 1-100 | 101-150 | 1-50| 51-150 |
> | Reminder     | 39.0         | 21.3   |41.6  | 19.2     |  36.1     | 16.4| 47.1|20.4 |
> |Ours| **48.0**| **34.8**| **49.1**|**36.8**| **46.7** | **27.5**  |**56.4** | **38.7**|
>
>    - **Improved Plasticity:** Our initialization step improves the model's ability to learn the new class. The model can distinguish features of the new class, rather than learning all features from scratch.
>    - **Improved Rigidity:** Because the new classifier starts from a well-informed position, the optimization process is far more stable. This in turn reduces the representational drift that causes catastrophic forgetting of old classes.
>
> As shown in our quantitative comparisons above, SELECT consistently and significantly outperforms REMINDER, particularly in longer task sequences with large number of classes.
>
> - **Our Core Contribution**
> Therefore, our claim—that "current methods often fail to leverage prior knowledge effectively"—is specifically referring to use inter-class foreground relationships as an initialization strategy.
>    - **REMINDER** uses this knowledge for a loss function, not initialization.
>    - **MBS** uses initialization, but from the background class, not from similar foreground classes.
>
> *We apologize for not including REMINDER in our analysis. We have now updated Section 3.2 to include the work.*

---

> ### Author Response · Authors · 2025-11-24
> **Weaknesses 2: Impact of perturbation on past knowledge (Part 2 of 5)**
>
> ### Degradation in the performance of past knowledge
> As mentioned in Incrementer [1], if the classes in the incremental task exhibit semantic similarity with the previous task, the performance of these similar classes drops significantly. We observed a similar trend in Table 3 (in the paper) as well.
>
> ### Formulation of the Gap:
> The perturbation is formally defined as:
>
> $\hat{\theta_{CTA}} = \alpha \theta_{CTA} + (1 - \alpha) \mathcal{N}(\theta_{CTA}, \sigma^2)$
>
> where $\theta_{CTA}$ is the aggregated representation of prior similar classes.
> - **Without Perturbation ($\alpha=1.0$):** The new class initialization $\hat{\theta}_{CTA}$ lies exactly on the linear subspace spanned by the prior similar classes.
> - **With Perturbation ($\alpha < 1.0$):** The noise term $\mathcal{N}$ introduces a stochastic shift. This is the "gap."
>
> ### Preventing Knowledge Loss
> - **Without Perturbation:** When we initialize a new class token (e.g., "Sheep") using a similar old class token (e.g., "Dog" or “Cat”), if we transfer the weights exactly (without perturbation), the model effectively starts with identical representations for both classes.
>    - **The Consequence:** The decision boundaries for "Sheep" and "Dog" completely overlap. The model cannot distinguish them, leading to immediate confusion where old "Dog" images are misclassified as the new "Sheep" class. To correct this, the optimizer has to drastically alter the weights to separate them, which damages the original "Dog" knowledge (Catastrophic Forgetting).
>
> - **With Perturbation:** The perturbation creates a distinct initial identity. The new class starts in a region of the feature space that is semantically related but spatially distinct.
>    - **The Protection:** The gradient updates for the new class are directed toward refining this specific, noisy initialization. This enables the model to adapt to the new class without altering the decision boundaries of the existing classes.
>
> ### How this Prevents Knowledge Loss:
> This "gap" protects past knowledge in two ways:
> - **Reduced Interference:** Since the new class begins with a distinct weight vector, the gradients calculated to minimise the loss for "Sheep" do not directly conflict with the preserved weights of "Dog." The model doesn't have to unlearn Dog to learn Sheep; it just has to refine the distinct Sheep initialization.
> - **Guided Flexibility:** As mentioned in the draft, the perturbation provides flexibility. It moves the initialization into a high-probability region for the new class. This allows the model to refine the new class using new images without overwriting the precise boundary of the old class.
>
> ### Empirical Validation:
> Our ablation study provides direct evidence of this.
> - This theoretical explanation is backed by our results in Table 3. A brief table is presented below. We observed that without perturbation, the performance on previous similar classes dropped significantly because the model treated the new class as a duplicate of the old one, causing severe interference. The perturbation restores this balance.
>
> | Classes | w/o Perturbation | w Perturbation |
> | -------- | -------- | -------- |
> | Aero     | *16.0*     | ***93.7***     |
> | Bike     |  44.7     | 49.1     |
> | bird     |  87.6     |   90.3     |
> | boat     |  80.1     |  81.4     |
> | bottle     |  78.9     | 87.8     |
> | bus     |  92.4     |  95.2     |
> | car     |  88.5     | 92.0     |
> | cat     |  94.4     |   95.3     |
> | chair     |  50.2     | 51.9     |
> | cow     |  *48.5*     |  ***90.9***     |
> | table     |  63.2     |  64.1     |
> | dog     |  92.5     | 94.2     |
> | horse     |  90.7     |   91.3     |
> | motor     |  89.3     |   90.7     |
> | person     |  88.7     |  89.4     |
> | plant     |  72.1     | 68.2     |
> | sheep     |  *9.3*     | ***89.0***     |
> | sofa     |  44.5     |   49.6     |
> | train     |  *42.1*     |  ***86.9***     |
> | tv     |  57.7      |  67.3     |
>
> [1] Shang, Chao, et al. "Incrementer: Transformer for class-incremental semantic segmentation with knowledge distillation focusing on old class." Proceedings of the IEEE/CVF Conference on Computer Vision and Pattern Recognition. 2023.

---

> ### Author Response · Authors · 2025-11-24
> **Weaknesses 3: Context Transfer Loss vs MBS orthogonality loss (Part 3 of 5)**
>
> While both our **Context Transfer Loss (CTL)** and the **MBS orthogonality loss** involve separating class representations, the comparison is misleading as our objective and mechanism are fundamentally different. The core novelty lies in the objective of our entire framework, which our loss is specifically designed to serve.
>
> ### In the MBS orthogonality loss:
> - It forces the new class (e.g., "sheep"), which is randomly initialized, to become orthogonal to all existing classes (e.g., "dog," "cat," "car," "tree"), regardless of their semantic characteristics.
> - It actively tries to make "sheep" just as dissimilar to "dog" as it is to "car". This is counterintuitive and forces the model to discard any shared visual features.
>
> ### In our Context Transfer Loss (CTL):
> - Embracing Similarity: We first initialize the new class ("sheep") by transferring knowledge from its most similar old classes ("dog" or “cat”). At this point, the "sheep" and "dog" (or “cat”) representations are intentionally close, sharing a common foundation.
> - Context Transfer Loss (Guided Separation): Further, we apply our CTL. As we stated, its goal is to "encourage distinct information between previously learned similar classes and the transferred knowledge."
>    - Its sole job is to gently push "sheep" and "dog" apart just enough so the model can distinguish them, without degrading the shared knowledge.
>
> ### Experiments to Validate This Distinction
>
> To validate this distinction, we present two distinct evaluations:
>
> - Experiment 1: Validating the Problem
>    - SELECT + MBS Orthogonality Loss
>    - SELECT + CTL loss
>
> | Model Variant | 15-5 |  |
> | -------- | -------- | -------- |
> | SELECT + MBS Orthogonality Loss     | 82.3| 73.0
> | **SELECT + CTL**     | **83.9** | **76.0**|
>
> - Experiment 2: Validating our context transfer loss
>    - SELECT + No CTL loss
>    - SELECT + CTL Loss
>
> | Model Variant | 15-5 |  |
> | -------- | -------- | -------- |
> | SELECT + No CTL     |83.2| 70.2 |
> | **SELECT + CTL**     | **83.9** | **76.0**|

---

> ### Author Response · Authors · 2025-11-24
> **Weaknesses 6: Interpretation of Table 4 (Part 4 of 5)**
>
> We respectfully disagree with the interpretation that the contribution of these losses is minimal. This ablation study demonstrates the absolute necessity of our proposed losses for the ability to learn new classes.
>
> We would like to bring the reviewer's attention to the catastrophic forgetting on the 15-1 task.
> ### The Catastrophic forgetting on the 15-1 Task
> - In the 15-1 split, when the contrastive and feature distillation losses are removed, the model's ability to learn the new class completely collapses.
> - A drop in new-class mIoU from 72.0% to 24.4% demonstrates that they are the primary losses enabling the model to learn new concepts. The overall mIoU drop from 80.5% to 66.8% confirms this.
> ### Validating 15-5 Result
> - The reviewer correctly notes the 'all' mIoU on 15-5 increases slightly from 81.6% (Ours) to 82.1% (Ablated). However, a closer look at why this happens reveals that:
> - The ablated model performs poorly at learning new classes, with a 2.3% drop in performance.
> - The ablated model exhibits a slight improvement in retaining old classes, with a minor gain of +0.2%.
>
> In the 15-5 split, the 'all' mIoU is a weighted average and biased towards the 15 old classes. The +0.2% gain on 15 classes outweighs the -2.3% loss on 5 new classes, resulting in an increase in the overall average of the ablated model..

---

> ### Author Response · Authors · 2025-11-26
> **Weaknesses 5: Cost of Training (Part 5 of 5)**
>
> We respectfully suggest to the reviewer that focusing solely on the "All" mIoU metric provides an incomplete picture of the method's contribution and efficiency. Below, we address the performance gain and the cost analysis:
> ### Re-evaluating the performance gain:
> The "All" mIoU metric is a weighted average that is heavily biased toward the large number of base classes (e.g., 100 base classes vs. 5 new classes).
> - While the "All" gain appears small, our method achieves substantial gains over new classes.
> - For example, in the ADE20K 100-50 and 50-50 task, while the "All" gain might seem moderate, the New Class gain is significant compared to baselines.
> - The primary challenge is learning new concepts without catastrophic forgetting. A method that improves "New" class learning by a large margin is highly valuable, even if the "All" average (diluted by old classes) moves slowly.
>
> ### Complexity & Cost Analysis:
> - Training Time: SELECT uses a "prior" initialization. Because the new class weights start in a semantically relevant space, the model converges faster than methods starting from background initialization or methods that need to build prototypes from scratch (like NeST) (as seen in Fig. 1 in the paper).
>
> | Model | Training Time (hrs) |
> | -------- | -------- |
> | NeST     | 2.6     |
> | **Ours**     | **1.8**     |
>
> - Memory Cost: NeST and INC often require storing prototypes or additional attention parameters. SELECT only requires the storage of the Context Transfer Loss calculation, which is negligible.
>
> | Model | Incremental Step Training Parameter (in thousands)|
> | -------- | -------- |
> | NeST     | 5.4     |
> | **Ours**     | **0**     |

---

### Author Response · Authors · 2025-12-03
**Summary and Common Response**

### *Dear Area chairs and Reviewers*,

We appreciate your time and insightful feedback. We are encouraged that the reviewers recognized the novelty of our work, its strong motivation, and its effectiveness in identifying key weaknesses in prior methods. The feedback has been invaluable and has significantly strengthened the presentation and clarity of our work. We write this final remark to synthesize the contributions of our work, address all the feedback and concerns, and present the context for the final decision.

### **Paper Summary and Motivation**

This paper introduces a selective knowledge transfer strategy for class-incremental semantic segmentation (CISS) to address the challenges of stability, knowledge transfer and plasticity. The core idea is to identify semantically related classes from previously learned ones and selectively transfer knowledge to new classes. The central motivation for our work stems from a fundamental aspect of lifelong learning: the ability to solve new problems by relying on relevant past experiences. *Can previously learned knowledge create an impact in learning a new, unknown set of classes?*

Inspired by this, we apply this learning paradigm in continual semantic segmentation. Unlike prior works that leverage background knowledge, we present the experimental validation of the importance of adaptive learning over background knowledge. By building a foundational understanding of new classes from the most relevant old knowledge, our approach presents a more sophisticated learning process.

Our primary contributions are:

- **Class Similarity Detection:** A novel strategy that uses representation distance metrics to identify which previously learned classes are most semantically similar to a new class.
- **Context Transfer Attention (CTA):** An attention mechanism that adaptively transfers relevant information from the previously learned similar classes, providing an informed and robust initialization for the new class representations.
- **Context Transfer Loss ($\mathcal{L}_{ct}$):** A dedicated loss function that preserves the distinction between the newly initialized class representations and previously learned similar representations, and ensures that transferred knowledge does not degrade the learned representation of previous similar classes.

Beyond achieving promising performance across different datasets under diverse settings, including some challenging scenarios, our work broadly emphasizes the significance of selective knowledge transfer, a crucial capability for intelligent systems that are made to learn many skills, leveraging previous knowledge.

### **Review Process and Discussion**
In response to constructive feedback from the reviewers, we have diligently addressed all concerns raised during the review period, provided detailed clarifications, theoretical justifications, and new empirical evidence to support our claims.
Below are the individual summaries for each reviewer’s feedback.

- **Reviewer EQWq:** EQWq acknowledges the relevance of our proposed strategy for CISS and appreciates the clarity and presentation of our work. The reviewer raised concerns about the distinction with recent works and the justification for ablation studies. We provided detailed theoretical and empirical evidence to support the distinction and further justify ablation studies, with significant evidence. We have also included the changes in the updated manuscript.

- **Reviewer 7RG7:** 7RG7 appreciates our discussion on prior works and further our proposed strategy, including token selection, CTA, noise and margin loss. The reviewer further sought clarification on the robustness of the perturbation, along with more enhanced experimental results. We have clarified all concerns and incorporated their suggestions into the updated manuscript.

- **Reviewer ReFa:** ReFa appreciates the motivation of our work and finds the selective transfer approach intuitive. The reviewer further praises the structure of our work, along with the well-illustrated diagram and clarity in writing. The reviewer sought clarification on the comparison with prior works and suggested proper justification for assumptions and a more enhanced analysis. We provided detailed justification for the comparison with prior works and presented theoretical and empirical evidence to support our assumptions. We have also accommodated these changes in the updated manuscript.

*Remark:* We believe this paper makes a significant contribution to continual learning by demonstrating how to effectively and safely leverage prior knowledge for new tasks. We hope the technical advancements, their implications and the constructive review discussions will be carefully considered in the final decision.



Warm regards,

*Authors of #4783*

---

### Meta-Review · Area_Chair_4fjs · 2026-01-06

**Summary:**

This submission was reviewed by three expert reviewers, with the ratings of: 2 reject, and 1 borderline reject. The major concerns from the reviewers are about inaccurate claims, unclear presentations of the techniques, technical novelty compared to related works, the performance, background shift problem, unfair experimental comparison, inconsistency between the reproduced results and the original ones in the literature, unvalidated assumptions, and sensitive hyperparameters. The authors provided a rebuttal for the concerns, but no further response from the reviewer was presented, and there was no discussion during the rebuttal phase.

After carefully going through all the review comments and the rebuttal from the authors, it can be seen that some concerns are addressed by the rebuttal and further experiments. However, major concerns still remain not well addressed. There is no clear support for acceptance. As a result, it is unfortunate that this submission is not ready to be accepted to ICLR in its current form and needs another round of revision followed by reviews.

**Reviewer Concerns:**

Concerns that the AC thinks were addressed by the rebuttal: more details are provided for the unclear presentation, but still confusing; robustness of the token perturbation was partially addressed; performance on more architectures than ViT; hyperparameter selection; clarification on the background shift; unfair comparison with NeST; substantial reproduction gaps partially addressed by providing and confirming the experimental settings, though the issue still exist.

Concerns that are still outstanding: the misleading claims; unclear and missing details for the proposed techniques; technical contributions and novelty; lower performance; the issue with the contrastive loss was explained, but still unclear and unconvincing; the substantial reproduction gaps still exist without a convincing clarification; the invalidated core assumption concern, though more details are provided.

**Reviewer Scores:**

According to the review comments, and the rebuttal, for each review the reviewer might have changed their score in the way below, if they had been able to participate fully in the discussion:
* Reviewer EQWq: reject unchanged, or change strong reject
* Reviewer 7RG7: reject unchanged, or to borderline reject
* Reviewer ReFa: borderline reject unchanged, or to borderline accept.

---

### Decision · Program_Chairs · 2026-01-26

Reject